# A Mechanistic Analysis of Sim-and-Real Co-Training in Generative Robot Policies

**Yu Lei** [1]  **Minghuan Liu** [1]  **Abhiram Maddukuri** [1]  **Zhenyu Jiang** [2]  **Yuke Zhu** [1 3]

https://science-of-co-training.github.io/

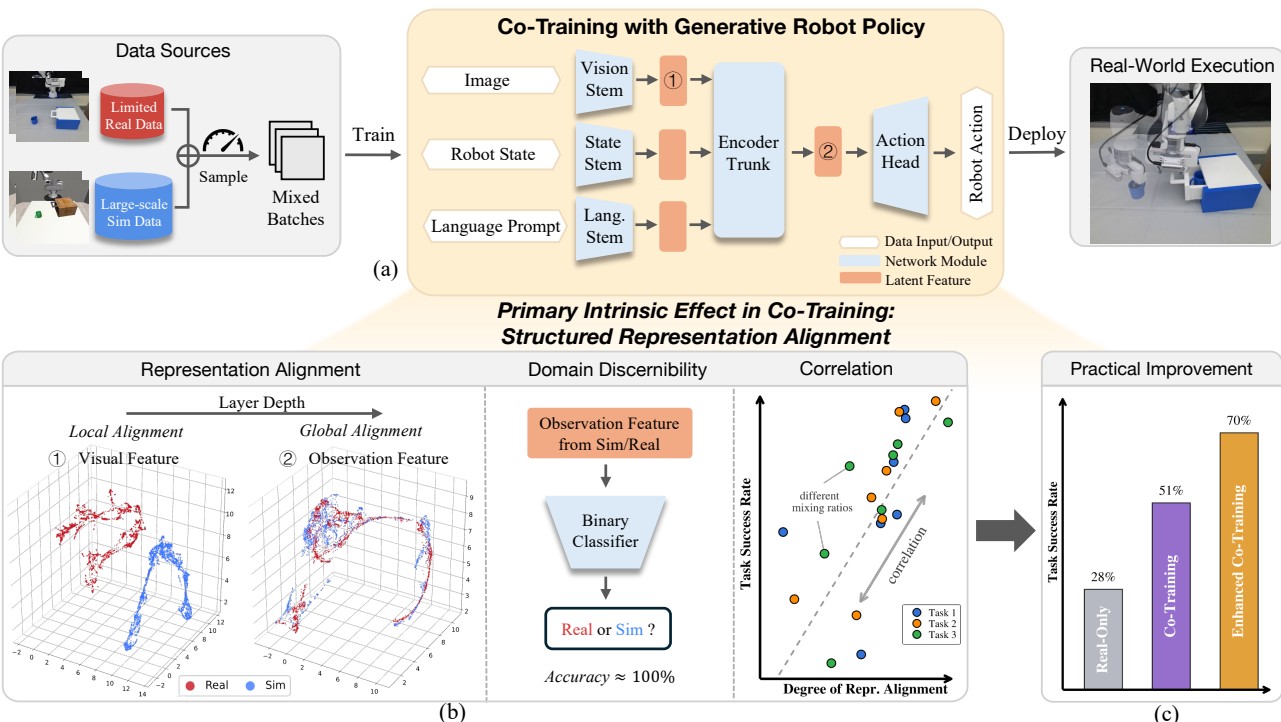

*Figure 1.* (a) **A workflow example of co-training systems** for generative robot policies. We identify *Structured Representation Alignment* as the main intrinsic effect in co-training: it refers to both representation alignment and domain discernibility, which lay the foundation of action transfer and adaptation. (b) **Representative observations**: representation alignment can be implicitly learned with appropriate mixing ratios, showing local geometric alignment in shallow layers and global alignment in deep layers (Sec. 4.1). The globally aligned features still preserve discernibility (Sec. 4.2). Representation alignment has a strong positive correlation with success rate. (c) Inspired by these observations, a simple fix of co-training with enhanced structured alignment can improve the success rate by another $\sim 20\%$.

## Abstract

Co-training, which combines limited in-domain real-world data with abundant surrogate data such as simulation or cross-embodiment robot data, is widely used for training generative robot policies. Despite its empirical success, the mechanisms that determine when and why co-training is effective remain poorly understood. We inves-

[1]Department of Computer Science, The University of Texas at Austin [2]Amazon FAR [3]NVIDIA. Correspondence to: Yu Lei <yulei@utexas.edu>.

*Proceedings of the $43^{rd}$ International Conference on Machine Learning*, Seoul, South Korea. PMLR 306, 2026. Copyright 2026 by the author(s).

tigate the mechanism of sim-and-real co-training through theoretical analysis and empirical study, and identify two intrinsic effects governing performance. The first, **"structured representation alignment"**, reflects a balance between cross-domain representation alignment and domain discernibility, and plays a primary role in downstream performance. The second, the **"importance reweighting effect"**, arises from domain-dependent modulation of action weighting and operates at a secondary level. We validate these effects with controlled experiments on a toy model and extensive sim-and-sim and sim-and-real robot manipulation experiments. Our analysis offers a unified interpretation of recent co-training tech-

niques and motivates a simple method that consistently improves upon prior approaches. More broadly, our aim is to examine the inner workings of co-training and to facilitate research in this direction.

## 1. Introduction

Data scarcity remains a fundamental bottleneck in robotics, motivating the use of inexpensive and abundant surrogate data such as simulation and cross-embodiment data (Bjorck et al., 2025; Physical Intelligence, 2025). Although these data sources contain rich task-relevant information, they introduce substantial domain gaps that make effective knowledge transfer difficult in practice. Recently, a simple *co-training* paradigm — jointly training on in-domain real data and surrogate data with a data mixing ratio $w$ — has demonstrated strong empirical performance across sim-and-real and human-to-robot settings (Cheng et al., 2025; Yuan et al., 2025; Kareer et al., 2025a;b). Despite some work (Wei et al., 2025) providing valuable empirical analysis, co-training remains poorly understood: its internal mechanisms are largely treated as a black box, and the factors that govern its effectiveness are unclear.

In this work, we focus on investigating the co-training paradigm as described above, in particular, on *sim-and-real* data, in diffusion-based models, which are representative in modern generative robot policies. (Pan et al., 2025)

We begin with a theoretical analysis that examines the learning objective induced by jointly mixing data from multiple domains. This analysis reveals two intrinsic effects that independently influence co-training performance: (1) **Structured representation alignment** is characterized by a two-fold property. On one hand, representations become aligned across domains in a domain-invariant subspace, enabling the transfer of task-relevant knowledge. On the other hand, representations retain discernibility with respect to domain-specific factors, allowing actions to adapt to the real world rather than being directly copied from surrogate domains. This balance is instrumental for effective co-training, as it determines whether adaptive action transfer is possible. (2) **Importance reweighting effect** refers to the domain-dependent logit modulation within action weightings. This effect operates locally in the space conditioned on observations and controls how much each training sample from each domain contributes to an action decision during training. It is determined by the data mixing ratio $w$, the dataset size $|\mathcal{D}|$ and domain gaps. Through controlled toy co-training experiments, we verify the presence of both effects and find that structured representation alignment is the primary factor underlying strong model performance, while the importance reweighting effect plays a modulatory role. These findings motivate the following questions:

*Do similar effects arise in realistic robot manipulation tasks? How can these insights guide the design of more effective co-training algorithms?*

In the second part of the paper, we address these questions through comprehensive sim-and-sim and sim-and-real robotic manipulation experiments. In end-to-end co-training systems, the data mixing ratio $w$ is typically the only explicit control variable, yet it simultaneously influences both internal effects. Empirically, we find that structured representation alignment, in both local and global space, can emerge implicitly within an appropriate range of mixing ratios (which we refer to as "balanced mixing ratios"), and that its strength exhibits a moderate-to-strong correlation with task success. At the same time, preserving domain discernibility is necessary for effective action adaptation to the real world; when this property is lost, performance even shows a negative correlation with representation alignment. These observations provide a unified perspective for understanding existing co-training techniques. We benchmark three recent representative co-training techniques on our tasks — optimal transport-based feature regularization (Cheng et al., 2025; Punamiya et al., 2025), adversarial discriminative domain adaptation (Cai et al., 2025; Yuan et al., 2025), and classifier-free guidance (Wei et al., 2025). We observe that each method primarily emphasizes only one aspect of structured representation alignment, which often leads to unstable or marginal improvements. Motivated by this analysis, we propose a simple combination of co-training techniques that jointly promotes alignment while preserving domain discernibility, and that also offers a more controllable interface for knowledge transfer during inference. This approach yields consistent and substantial improvements over prior methods. In summary, our contributions are as follows.

- We systematically identify, for the first time, the working mechanisms of co-training through theoretical analysis and experimental support.

- We find that structured representation alignment can be learned implicitly, and validate the effects and requirements of both alignment and discernibility through comprehensive robotic manipulation experiments.

- We benchmark representative co-training techniques through the lens of our analysis, which inspires us to introduce a simple approach that stably improves performance, and opens the door to new algorithm designs.

## 2. Theoretical Analysis of Co-Training

Generative robot policies use generative modeling architectures, such as diffusion/flow models, and autoregressive transformers, as parameterizations of the mapping from observation to action. Given their popularity and adoption

in industry, our analysis throughout this paper focuses on the most popular policy form — diffusion/flow matching policy (Chi et al., 2025). In the following, we use "diffusion" as an umbrella term for both diffusion and flow matching (Liu et al., 2022) models, as they are equivalent under our analysis (Gao et al., 2025). We start with illustrating why structured representation alignment matters in co-training with diffusion policy (Sec. 2.1); then, we provide an analysis of the importance reweighting effect (Sec. 2.2).

## 2.1. Structured Representation Alignment

Training diffusion policy corresponds to jointly learning a feature encoder $f_\phi : \mathcal{O} \to \mathcal{Z}$ to project observations into a latent space, and a policy model $\pi_\theta : \mathcal{Z} \to \mathcal{A}$ that maps the learned representations to the action space. Formally, given limited robotic dataset $\mathcal{D}_T = \{(o_i, a_i)\}_{i=1}^N$ in the target domain, and abundant dataset $\mathcal{D}_S = \{(o_j, a_j)\}_{j=1}^M$ from a source domain ($M \gg N$), we train a diffusion policy model (Chi et al., 2025) with mixing ratio $w$. This gives us the learning objective as:

$$\mathcal{L}_w(t; \phi, \theta) := w \cdot \mathcal{L}_{\mathcal{D}_T} + (1 - w) \cdot \mathcal{L}_{\mathcal{D}_S} \quad (1)$$

where $\mathcal{L}_\mathcal{D} = \mathbb{E}_{(o_i, a_i) \sim \mathcal{D}, \epsilon \in \mathcal{N}(\mathbf{0}, \mathbf{I}_d)}[||\epsilon - \epsilon_\theta(a^t, t, o)||_2^2]$. We can prove that there exists an analytical optimal solution. In this paper, we adopt the score parameterization (proof provided in Appendix B.1):

$$s_w^*(a^t, t, o) = \hat{w}_t \cdot s_t^*(a^t, t, o) + \hat{w}_s \cdot s_s^*(a^t, t, o) \quad (2)$$

where

$$\hat{w}_t = \frac{w \cdot p_t(a^t, f_\phi(o))}{w \cdot p_t(a^t, f_\phi(o)) + (1 - w) \cdot p_s(a^t, f_\phi(o))}$$

$$p_k(a^t, z) = \frac{1}{|\mathcal{D}_k|} \sum_{i \sim \mathcal{D}_k} p(a^t | a_i^0) \cdot K(z, z_i), \quad k \in \{t, s\}. \quad (3)$$

Here $K(\cdot, \cdot)$ is a kernel measuring how closely the current observation matches the observations in the dataset. So the behavior of the empirical optimal score function depends heavily on the learned observation representations $z = f_\phi(o)$. Based on the degree of representation alignment between the source and the target domain induced by $f_\phi$, we hypothesize three different scenarios in co-training: 1) **Disjoint:** observation representations of source and target domains are located in totally different clusters. During inference in the target domain, with $p_s(a^t, z) \approx 0$, the dynamic weight $\hat{w}_t$ stays near 1. So the policy ignores data from the source domain, thus no *positive transfer* from source to target will occur. 2) **Structured aligned:** the policy learns task-relevant, domain-invariant representations while retaining sufficient domain-specific information, such that observation representations of source and target domains are close but not collapsed. In this case, the action

prediction will be effectively guided by neighbors in the source domain but dominated by data from the target domain. This informs our definition of structured representation alignment at the beginning. 3) **Overlapping:** Although the observation representations of the source and target domains are fully aligned, the corresponding actions differ due to domain gaps. As a result, the policy prediction is unaware of the actual environment and instead exhibits a bimodal distribution over source and target actions, leading to *negative transfer*.

## 2.2. Importance Reweighting Effect

Mixing ratio directly provides additional modulation in this transfer process. Given any specific observation $o$, Eq. (3) will degrade to:

$$\hat{w}_t := \frac{w p_t(a^t)}{w p_t(a^t) + (1 - w) p_s(a^t)} \quad (4)$$

where $p_t(a^t) = \frac{1}{N} \sum_{i:o_i=o}^N p(a^t | a^0 = a_i), p_s(a^t) = \frac{1}{M} \sum_{j:o_j=o}^M p(a^t | a^0 = a_j)$. At a large $t$ during inference, as data are greatly perturbed by noise, $p_t(a^t) \approx p_s(a^t)$, the model will approximate a global average between two domains; at a smaller $t$, $p(a^t)$ will be concentrated on one domain with the existence of domain gaps, so the model prediction will converge to one specific domain with $\hat{w}_t \approx 1$. At any timestep $t$, as $a^t$ are usually distributed as Gaussians around the training sample, for each data point we define: $r_k(a^t, t) := \frac{||a^t - \alpha_t a_k||}{\sigma_t \sqrt{d}}$. We can have a closer look at the merged score function with further simplifications:

$$s_w^*(a^t, t) = \sum_{i_t}^N g_{i_t} s_{i_t}^* + \sum_{i_s}^M g_{i_s} s_{i_s}^*$$

$$g_{i_k} = Softmax(\ln(w_k) - r_k^2(a^t, t) * d/2), \quad k \in \{t, s\}$$

$$w_t = w/N, w_s = (1 - w)/M$$

$$(5)$$

where $s_i^*$ is the optimal score towards each action data (derivation provided in Appendix B.2). It reshapes the action sampling distribution to learn by reweighting the score functions during training as shown in Fig. 10.

In a special case, we can further have the following relation of the relative weight ratio:

$$\frac{g_{i_t}}{g_{i_s}} \propto \mathcal{F}(\frac{w}{1 - w}, \frac{M}{N}, |a_{i_t} - a_{i_s}|) \quad (6)$$

The amplitude of this modulation is influenced by both $w$, the dataset size and domain gaps. A detailed characterization about this property is provided in Appendix B.3.

Based on the analysis above, we can summarize as follows: The effectiveness of co-training with diffusion policy is mainly decided by two intrinsic effects: (1) structured

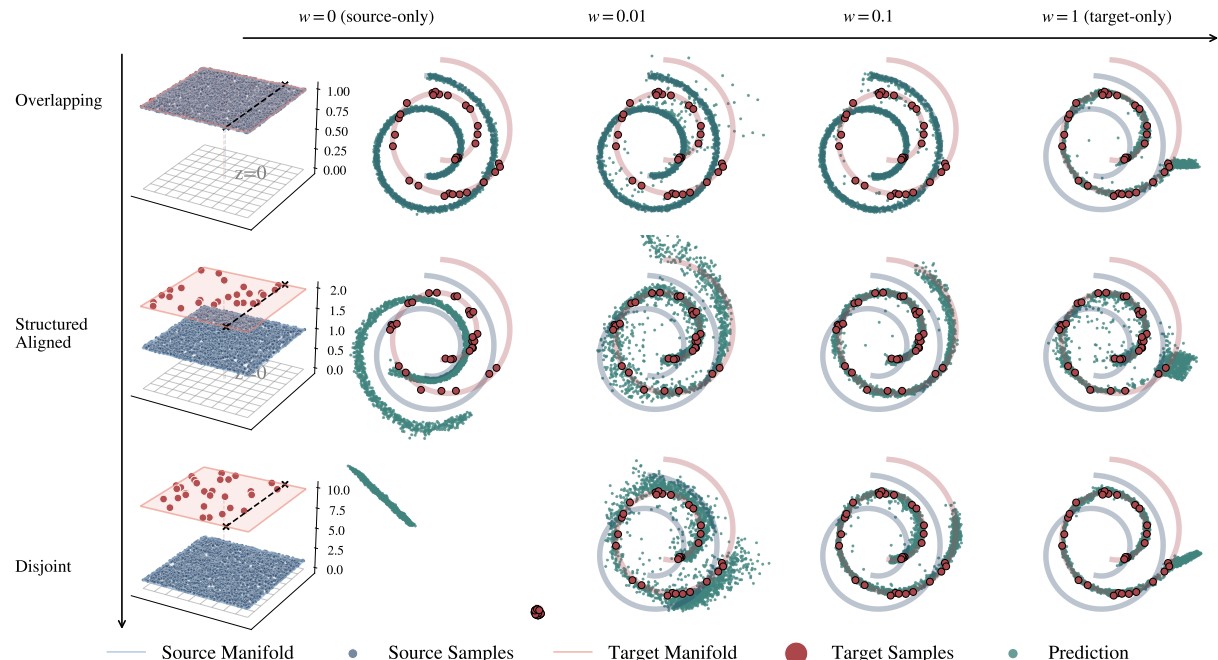

*Figure 2.* **Visualization of controlled toy example.** We co-train ∼30 and ∼3000 samples from target and source domain. Vertically in each column, the differences of prediction samples showcase the impact of representation alignment; horizontally in each row, the differences showcase the shift of importance reweighting effect controlled by mixing ratio $w$.

representation alignment; (2) the importance reweighting effect. With the above theoretical analysis, we are now ready to find empirical evidence that supports our insight.

## 3. Controlled Toy Example

Generally, the two effects interact during end-to-end co-training and jointly influence the learning dynamics. To disentangle and understand their individual contributions to co-training, we designed a pilot toy experiment. In this simplified setting, we seek to learn a policy model $\pi_\theta$ with a pre-trained feature encoder that defines the input distribution $p(x)$, where each input dimension corresponds to a principal direction in the latent space. We adopt a 4-layer Multi-Layer Perceptron (MLP) as the diffusion model architecture.

**Experiment design.** The policy model $\pi_\theta$ is co-trained to learn $\pi(y|x) : \mathbb{R}^3 \rightarrow \mathbb{R}^2$. We manually define two manifolds, $\mathcal{M}_S$ and $\mathcal{M}_T$, corresponding to the intrinsic data distributions of the source and target domains respectively. Then, we sample paired data points from these two manifolds $D_S = \{(x_i, y_i)\}^{N_S} \sim \mathcal{M}_S$ and $D_T = \{(x_i, y_i)\}^{N_T} \sim \mathcal{M}_T$, where $N_S \gg N_T$ and $D_T$ is sampled partially as shown in Fig. 2. This design simulates the common case in which target domain data is usually sparser and less diverse than source data. We align the two manifolds along two principal directions but vary the distance between them along the remaining one to create different representation scenarios. The results are shown in Fig. 2.

**Finding 1: The toy model behavior aligns with our theoretical analysis.** As expected, under three different representation scenarios described in Sec. 2.1, the co-trained model exhibits distinct behaviors: 1) In disjoint, the prediction remains close to that of the model trained solely on target-domain data, where the model can easily distinguish between two domains but fails to transfer knowledge, i.e., the learned mapping, from the source domain. Due to the limited amount of data, the model tends to memorize each data point (He et al., 2025), failing to interpolate within the training distribution and extrapolate beyond it. 2) In structured alignment, this setting represents a sweet spot, where the model achieves a balance between representation alignment and domain discernibility. As a result, the output distribution is reconstructed with high fidelity. 3) In overlapping, the model predictions become randomly distributed between the source and target domains. Here, the model cannot effectively distinguish between the two domains and instead treats them as identical, preventing meaningful adaptation of transferred knowledge. On the other hand, the data mixing ratio $w$ exerts an independent but secondary influence on this capability via importance reweighting, as discussed in Sec. 2.2. Specifically, it adjusts the relative amplitude of transferred knowledge $s_s^*$ and target-domain adaptation $s_t^*$. As illustrated by the horizontal comparison in Fig. 2, when $w$ is relatively small (e.g., the second column from the left), the output becomes noisier due to the increased contribution of source-domain data during the

early denoising steps.

In addition, we observe an intriguing phenomenon: with appropriate co-training settings, the model can reasonably make predictions in the out-of-distribution (OOD) region, indicating OOD generalization capability. Notably, this capability does not arise from simply copying knowledge from the source domain; rather, it emerges from preserving the distribution shift in the learned representations, which is crucial for accurate OOD prediction.

**Finding 2: Structured representation alignment is a dominant driver of strong model performance.** Since we have the ground-truth mapping, we provide a quantitative measure using $L2$ loss in Fig. 3. The overall importance reweighting effect is constrained by the underlying representation alignment. That is, changing the mixing ratio alone cannot compensate for poorly aligned representations, nor can it induce OOD generalization in the absence of sufficient alignment, *e.g.*, the red and blue curves where the mixing ratio nearly has no effect on the final performance. Based on this, we conduct an ANOVA-style variance decomposition analysis (Fisher & Fisher, 1971) on these two factors. We find that changes in structured representation alignment explain around 50% of the loss variance, while the importance reweighting effect of the mixing ratio accounts for only 20%. In this sense, structured representation alignment is the primary determinant of model behavior, while $w$ serves as a modulation factor that fine-tunes the balance between source and target domain knowledge.

By drawing an analogy from the toy example to sim-and-real co-training, we hypothesize a similar underlying mechanism: structured representation alignment enables effective knowledge transfer from simulation, while maintaining sufficient domain discernibility to adapt actions to the real world. A key question, however, is whether this mechanism can be empirically observed in practical sim-and-real settings, rather than remaining a purely conceptual intuition.

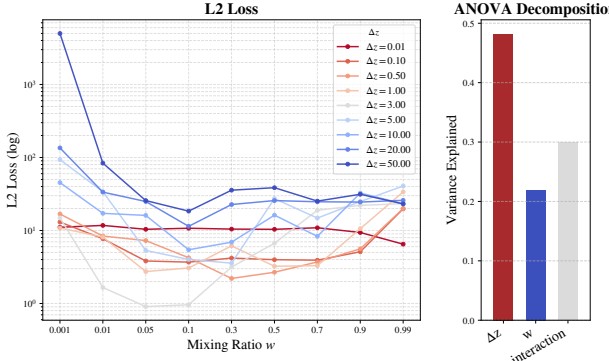

*Figure 3.* **L2 loss with sweeping mixing ratio and delta $z$ distance, and ANOVA variance decomposition.** When representations overlap (red line) or are disjoint (blue line), co-trained models are insensitive to mixing ratios. The importance reweighting effect (blue bar) can only explain 20% of the performance variance.

Moreover, *Finding 2* raises a deeper question: can structured representation alignment emerge in end-to-end co-training, given that the data mixing ratio $w$ is the only explicit control variable? To answer these questions, we conduct extensive experiments on real-world robotic manipulation tasks.

## 4. Sim-and-Real Co-Training for Manipulation

To find out more empirical evidence to validate our hypothesis in robot manipulation, in particular, sim-and-real co-training, we design a set of sim-and-sim and sim-and-real co-training experiments on manipulation tasks (Fig. 9). The sim-and-sim experiments are designed to explicitly control the domain gaps between source and target domains to ensure our observations are consistent across different domain gaps. Across all experiments, we adopt a transformer-based diffusion model (Chi et al., 2025) with ResNet18 (He et al., 2016) as the vision backbone and train end-to-end.

**Task suites.** We choose three manipulation tasks from robosuite (Zhu et al., 2020): `NutAssembly`, `MugHang`, and `MugCleanup`. `NutAssembly` and `MugHang` require more precise control than common pick-and-place tasks, as it includes dense object interactions. And there are more rotation motions in the actions of `MugHang` demonstrations. In addition, the model needs relatively long-horizon reasoning and execution to succeed in `MugCleanup`. These tasks represent many key challenges in robot manipulation.

**Environment setup.** For sim-and-real experiments, following the recipe in Maddukuri et al. (2025), we calibrate the camera pose and intrinsics to minimize camera alignment differences between simulation and the real world. For sim-and-sim experiments, we utilize the same *source-sim* environments and create a second *target-sim* environment with domain gaps.

**Domain gaps categorization.** Simulation and the real-world data contain domain shifts from various aspects. To identify the different effects of co-training across them, we decompose the gaps along two dimensions — visual appearance and environment physics. We manually introduce these gaps and construct three sim-and-sim co-training settings, *i.e.*, *visual-only*, *physics-only* and *visual-physics*.

**Data preparation.** For the target domains, we collect 50 human demonstrations for each task. For the source domains, we use MimicGen (Mandlekar et al., 2023) to further synthesize ∼3000 trajectories based on 50 human demonstrations. Following Wei et al. (2025), we define $w_n = \frac{|D_r|}{|D_r|+|D_s|}$ as the natural mixing ratio, where $|D_r|$ and $|D_s|$ are the sizes of real-world and simulation datasets, respectively. This is equivalent to concatenating the sim and real datasets. For experiments in this section, we co-train policies by sweeping a set of mixing ratios $w \in \{0, 0.005, w_n = 0.016, 0.1, 0.3, 0.5, 0.8, 1\}$.

### 4.1. Observations on Representation Alignment

**Representation alignment can be learned *implicitly* in end-to-end co-training.** Our experiments began with visualizing the latent embeddings of simulation and real-world observation features across different layers using UMAP (McInnes et al., 2018) at different mixing ratios. Specifically, we look into the features after the vision stem and the final-layer output embeddings of encoder trunk $f_\phi$, which include other modality information such as proprioception and language. What is surprising is that, in a certain range of mixing ratios, visual features exhibit local geometry alignment sharing very similar geometric structures, while the observation features show representation alignment in global space, as shown in Fig. 4. This can inform us about how the representation alignment evolves through the networks. We further quantify the local and global representation alignment using the Gromov-Wasserstein distance (Mémoli, 2011) and Wasserstein distance (Rüschendorf, 1985), respectively. By varying the data mixing ratio, we observe a clear correlation: smaller distances between real and simulation features correspond to more similar latent geometries and stronger alignment as shown in Fig. 4 (Full visualizations are available in Appendix D.2). This trend holds consistently across both sim-to-real and sim-to-sim experiments. These results suggest that co-training remains sensitive to the data mixing ratio $w$ because adjusting it simultaneously and substantially alters primary intrinsic effect—representation alignment itself. In other words, the mixing ratio does not merely re-weight source and target data contributions, but also implicitly re-shapes the learned representation space. Similar phenomena have also been observed in Kareer et al. (2025b), where the alignment emerges from scaling pre-training data.

**Representation alignment *positively correlates* with model performance.** We compute the correlation between the log-transformed Wasserstein distance obtained above and the corresponding success rate across different settings. For each checkpoint, we evaluate the policy over 200 trials (sim) and 30 trials (real) for computing the mean success rate. We report both Pearson's correlation coefficient, which captures linear associations, and Spearman's rank correlation coefficient, which is robust to non-linear but monotonic relationships. As shown in Fig. 5, in all settings except the physics-only condition in sim-and-sim co-training, both Pearson and Spearman correlation coefficients fall in the range of $0.6 \sim 0.8$, with p-values $< 0.04$. These results indicate a moderate-to-strong positive association between representation alignment and model performance. In some cases, one of the correlation coefficients (Pearson or Spearman) is lower (e.g., $\sim 0.4$), suggesting that the relationship may be non-linear or only partially monotonic rather than strictly linear. Importantly, this overall pattern is consistently observed across all three tasks.

**Suppressing representation alignment results in performance degradation.** To further verify the causal effect of representation alignment, we conduct a minimal ablation that explicitly encourages representation separation in *vis-phys* sim-and-sim setting. Inspired by adversarial domain adaptation (Tzeng et al., 2017), we keep the domain classifier operating on the learned representation, but intentionally remove the gradient reversal layer, thereby promoting domain-discriminative features instead of domain-invariant ones. The performance across 3 tasks drops consistently as shown in Table. 3.

### 4.2. Observations on Domain Discernibility

**Although representations align in low-dimensional space, they are easily discernible by shallow neural networks.** We perform a simple linear probing study by training a 2-layer MLP for binary-domain classification on the encoder trunk's output embeddings. Surprisingly, even if the representations seem to be aligned well in low-dimensional space, a simple MLP can easily achieve $\sim 100\%$ success rate on validation sets in all settings. This suggests that representations are in the partially aligned scenario, and co-training policies do retain the domain-specific information.

**Discernibility is *indispensable* for actions to adapt to the target domain.** We report the success rate of each task in each setting in Fig. 15. We can find that in the four sim-and-sim settings, the success rate of the *physics-only* policy is even lower than the *vis-phys* policy on the task of `NutAssembly` and `MugCleanup`. As we largely change the object's physical parameters while keeping its visual appearance similar, it is harder for co-trained policies to distinguish between the two environments. Interestingly, from Fig. 5, we observe that the correlation between representation alignment and model performance in the *physics-only* policy can even become negative, suggesting that blind representation alignment can be harmful.

## 5. A Unified View of Co-Training Methods

Although a wide range of co-training techniques has been proposed, it often remains unclear why these methods yield performance gains in some settings while failing in others. In this section, we revisit three representative co-training approaches through the lens of our findings and show that their empirical behavior can largely be explained by how they balance representation alignment and domain discernibility. Specifically, we categorize existing methods based on whether they primarily promote cross-domain alignment or preserve domain-specific information.

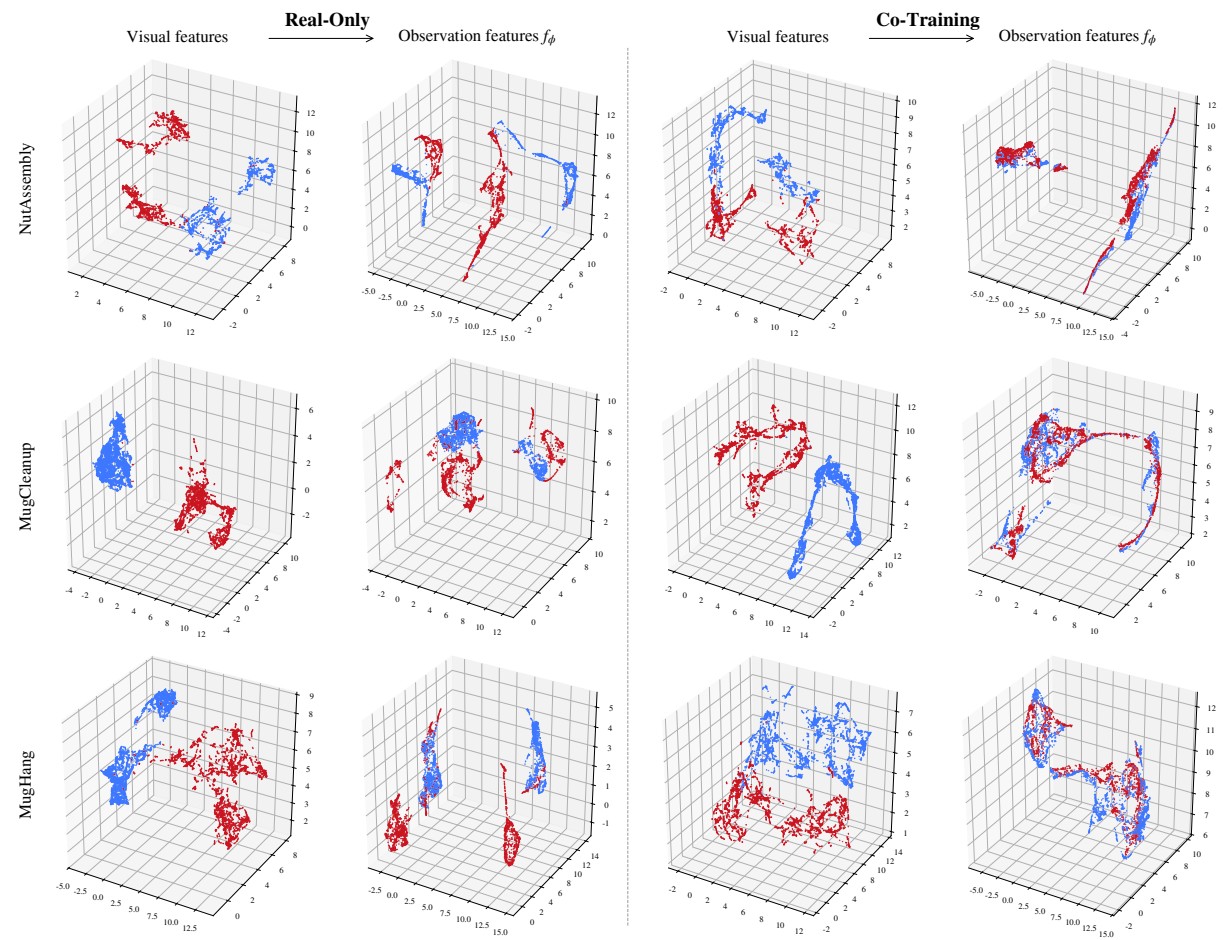

*Figure 4.* **UMAP visualization of latent features.** We visualize the features after the vision stem and after the encoder trunk $f_\phi$. Red and blue dots represent the real and simulation features, respectively. We show the results for a specific mixing ratio in this figure. More details and results are provided in the Appendix D.2.

### 5.1. Prior Co-Training Methods

**Optimal transport (OT)** (Courty et al., 2016)-based methods aim to align simulation and real-world data by explicitly matching their representation distributions, either in latent or trajectory space. Recent work (Punamiya et al., 2025; Cheng et al., 2025) formulated co-training as a joint optimal transport problem, in which samples from simulation and real domains are softly coupled to minimize a global discrepancy:

$$\min_{\phi,\theta} L_w(\phi,\theta) + \lambda \cdot L_{OT}(D_r, D_s). \tag{7}$$

$L_{OT}$ is usually computed using the Wasserstein distance between two domains. Under our hypothesis, such methods strongly encourage representation overlap across domains, effectively pushing simulation and real observations into a shared latent space. We implement OT-regularized co-training as in Cheng et al. (2025), and the only difference is that we drop the offline data pairing sampler.

**Adversarial domain adaptation (ADDA)** methods (Tzeng et al., 2017; Cai et al., 2025) similarly seek domain-invariant representations by training a discriminator to distinguish between simulation and real data, while simultaneously learning an encoder that attempts to fool the discriminator:

$$\min_{\phi,\theta} L_w(\phi,\theta) + \lambda \cdot L_{disc}(D_r, D_s). \tag{8}$$

$L_{disc}$ can be implemented simply using binary cross-entropy loss. From the perspective of our hypothesis, adversarial alignment also prioritizes cross-domain overlap, but does so implicitly through representation indistinguishability rather than explicit distribution matching. We implement it in the same way as Tzeng et al. (2017).

**Classifier-free guidance (CFG)** (Ho & Salimans, 2022) introduces a distinct mechanism for co-training by interpolating between conditioned and unconditioned policies at inference time. Rather than enforcing representation alignment during training, CFG modulates the influence of real-derived signals via a guidance scale. Under our hypothesis, CFG preserves domain discernibility by maintaining sepa-

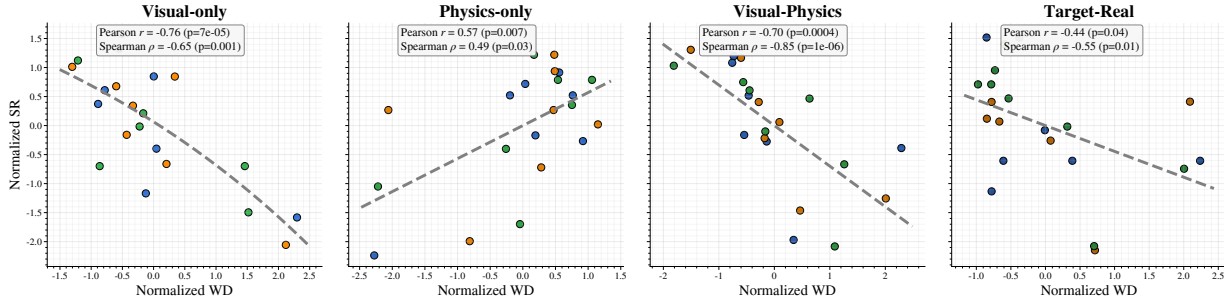

*Figure 5.* **Correlation between representation alignment and success rate.** We group the results of different tasks in each setting and compute the Spearman and Pearson correlation coefficients after normalization. Blue, yellow, and green-series dots represent the same task in Fig. 15. Gray dash line indicates the fitted curve.

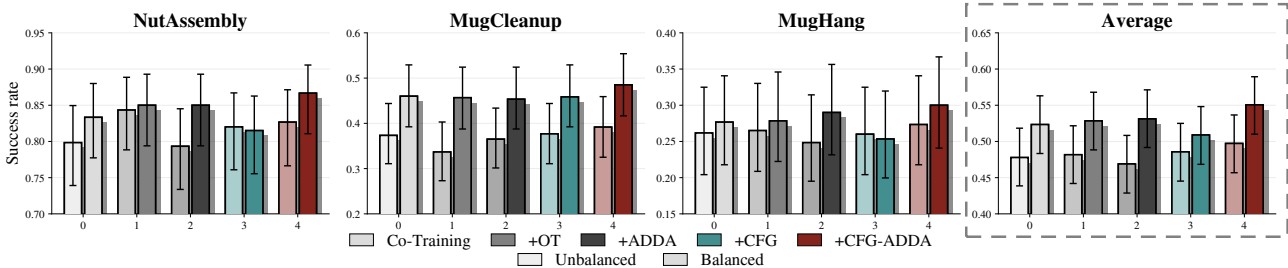

*Figure 6.* **Average performance on sim-and-sim experiments.** We can find that alignment-based methods (+OT/+ADDA) get improvement in the balanced mixing ratio group, while discernibility-based methods (+CFG) get improvement mainly in the unbalanced mixing ratio group. More detailed results are available in the Appendix. D.1

rate conditional pathways, while still enabling controlled knowledge transfer from simulation:

$$\tilde{s}_\theta(a, o, c, t) = (1+\lambda) \cdot s_\theta(a, o, c, t) - \lambda \cdot s_\theta(a, o, \varnothing, t). \quad (9)$$

We implement it by concatenating a one-hot embedding $c$ as environment labels to the observation features after the vision encoder, setting $\lambda$ to 0 as recommended in Wei et al. (2025).

**CFG-ADDA: a simple combination.** Viewed through our explanation framework, existing co-training methods primarily differ in how they trade off representation alignment and domain discernibility. OT and ADDA-based methods emphasize alignment, which can be beneficial when domain discrepancies are small but may lead to negative transfer when discrepancies are large. In contrast, classifier-free guidance preserves domain awareness while allowing flexible information sharing. This unified perspective clarifies the strengths and limitations of prior approaches and motivates our proposed combination strategy, which explicitly balances these two competing objectives. We simply combine the techniques of *CFG* and *ADDA*, named as *CFG-ADDA*. Specifically, we attach one-hot embeddings as environment labels to enable domain guidance, while encouraging the remaining representation dimensions to align through an adversarial discriminator. Training details are provided in Appendix C.

With this explicit disentanglement of domain-invariant and domain-specific features, we want to point out a new perspective towards the score interpolation coefficient $\lambda$ — as only the environment labels are dropped in $s_\theta(a, o, \varnothing, t)$, it actually represents the average log-probability gradient direction in all domains. So $\lambda$ can be viewed as a more flexible control variable to transfer the "averaged knowledge" during inference as opposed to transferring during training by importance reweighting effect. We set $\lambda = -0.5$ for *CFG-ADDA* as default.

### 5.2. Experiments and Analysis

**Sim-and-Sim Experiments.** We implement the above techniques on top of our co-training model and conduct visual-physics sim-and-sim co-training experiments. The results are shown in Fig. 6. We group the data mixing ratios into two regimes: balanced and unbalanced mixing. Performances with balanced mixing ratios are consistently better than with unbalanced mixing ratios. Under balanced mixing, where simulation and real data are present in comparable proportions, alignment-oriented methods (OT and ADDA) consistently improve performance across tasks. This indicates that representation alignment effectively facilitates cross-domain knowledge transfer when both domains are sufficiently observed during training. In contrast, under unbalanced mixing, alignment-only methods

| Method | Success Rate | | | | | | | | | Avg |
|---|---|---|---|---|---|---|---|---|---|---|
| | NutAssembly | | | MugCleanup | | | MugHang | | | |
| Real-only | 11/30 | | | 8/30 | | | 6/30 | | | 8.6/30 |
| Mixing Ratio $w$ | 0.016 | 0.1 | 0.3 | 0.016 | 0.1 | 0.3 | 0.016 | 0.1 | 0.3 | - |
| Co-Training | **17/30** | 11/30 | 16/30 | **16/30** | 9/30 | 7/30 | 8/30 | **13/30** | 7/30 | 15.3/30 |
| + OT | 15/30 | **17/30** | 11/30 | 8/30 | 15/30 | **15/30** | **11/30** | 9/30 | 4/30 | 14.3/30 |
| + ADDA | **13/30** | 13/30 | 15/30 | 6/30 | **14/30** | 11/30 | 10/30 | **14/30** | 7/30 | 14.3/30 |
| + CFG | **15/30** | 14/30 | 11/30 | 6/30 | **17/30** | 14/30 | 8/30 | **14/30** | 10/30 | 15.3/30 |
| **+ CFG-ADDA($\lambda = 0$)** | **20/30** | 17/30 | 11/30 | 15/30 | **19/30** | 14/30 | 15/30 | **17/30** | 13/30 | **18.6/30** |
| **+ CFG-ADDA($\lambda = -0.5$)** | **23/30** | 15/30 | 18/30 | 11/30 | **22/30** | 17/30 | **18/30** | 15/30 | 8/30 | **21/30** |

*Table 1.* **Real-world policy performance under different co-training strategies.**

exhibit pronounced performance degradation, particularly on `MugCleanup` and `MugHang`. This behavior suggests that when one domain dominates the training data, enforcing strong alignment biases the learned representation toward suboptimal invariances, thereby hindering real-world adaptation. CFG, which explicitly preserves domain information, demonstrates greater robustness in this regime, but its peak performance remains limited. Notably, CFG-ADDA achieves strong performance across both regimes. By combining adversarial alignment with explicit domain conditioning, it leverages transferable structure from simulation under balanced mixing while maintaining domain discernibility under unbalanced mixing.

**Sim-and-Real Experiments.** Since balanced mixing ratios are the main choice for effective co-training, we conduct real-world evaluations only in this regime. The observations in the Table 1 are similar to sim-and-sim settings. We even find more stable and substantial improvement with our proposed method in the real world, achieving $\sim 74\%$ success rate on these challenging tasks.

These results support our findings that effective sim-and-real co-training requires both representation alignment for transfer and domain discernibility for adaptive behavior.

## 6. Discussions and Future Work

To understand how co-training works, we present a systematic study that combines theoretical analysis with extensive empirical validation, yielding a unified explanatory framework. Within this framework, we identify two intrinsic effects underlying effective co-training: structured representation alignment and importance reweighting. The effectiveness of structured representation alignment requires a careful balance between two competing objectives: aligning representations along domain-invariant dimensions to enable transfer, while preserving domain-specific dimensions to maintain adaptability. This perspective unifies several existing co-training methods and highlights the effectiveness of a simple combination strategy. We further identify the

effects of mixing ratios and dataset size, which can help narrow the search space for future large-scale co-training experiments. A guideline is provided in Appendix D.5. Overall, we hope that this work sheds light on the mechanisms behind co-training and informs the design of more principled, robust co-training algorithms.

**Limitations and Future Work.** First, our empirical study primarily focuses on sim-to-sim and sim-to-real co-training settings. Although we observe qualitatively similar trends in other co-training scenarios, such as human–robot co-training, validating the generality of our findings across a broader range of domains remains an important direction for future work. Second, our analysis concentrates on the end effects of the two identified mechanisms, without explicitly characterizing their interaction during the dynamic learning process — particularly how the mixing ratio shapes representation learning over the course of training. In addition, we do not investigate the potential impact of practical factors such as limited batch sizes. Third, we study the relative relationships between representations, rather than their intrinsic structure. That is, we do not directly characterize what representations the model ultimately learns. Understanding the nature of these representations, especially those that generalize across domains, may provide further insights into the emergence of structured representation alignment. Finally, while our study is grounded in imitation learning, the co-training paradigm is broadly applicable to other learning settings, including world modeling and reinforcement learning. We hope this work encourages further exploration across diverse domains, ultimately contributing to a deeper understanding and more effective use of co-training.

## Acknowledgment

We would like to thank the Texas Advanced Computing Center (TACC) for its valuable support of computing resources. We also thank Rutav Shah, Huihan Liu, Kevin Lin, and Jake Grigsby at UT Austin Robot Perception and Learning Lab for their fruitful discussions. This work was partially supported by the National Science Foundation (FRR-2145283,

EFRI-2318065), the Office of Naval Research (N00014-24-1-2550), the DARPA TIAMAT program (HR0011-24-9-0428), and the Army Research Lab (W911NF-25-1- 0065). It was also supported by the Institute of Information & Communications Technology Planning & Evaluation (IITP) grant funded by the Korean Government (MSIT) (No. RS-2024-00457882, National AI Research Lab Project).

## Impact Statement

This paper presents work where the goal is to advance the fields of machine learning and robotics. There are many potential societal consequences of our work, none of which we feel must be specifically highlighted here.

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

# Contents

# A. Related Work

## A.1. Co-Training for Robot Learning

Generally, we can define any learning system that utilizes heterogeneous data as co-training. To mitigate the gap between limited in-domain robot data and large-scale surrogate data or even multimodal resources, co-training has been employed in numerous studies. Based on the types of large-scale surrogate data, we can coarsely categorize current work into the following three intersecting types: *sim-and-real co-training*, *cross-embodiment co-training*, and *non-robot data co-training*.

**Sim-and-Real Co-Training.** Using simulation as a surrogate data source is a promising way for co-training. The main domain gaps include visual and physics gaps, with the physics gap being much more challenging. But with the advancement of high-fidelity physics simulators (Mittal et al., 2025; Nasiriany et al., 2024; Zakka et al., 2025) and automated data generation tools (Mandlekar et al., 2023; Jiang et al., 2025; Lin et al., 2025), high-quality and massive robot trajectories can be obtained easily. Many works have demonstrated the effectiveness of sim-and-real co-training on challenging manipulation tasks with small diffusion policy models (Maddukuri et al., 2025; Wei et al., 2025; Cheng et al., 2025) and even large Vision-Language-Action(VLA) models (Bjorck et al., 2025; Yu et al., 2025). There are also works (Barreiros et al., 2025; Jain et al., 2025) utilizing relatively large-scale real-world data and a small amount of simulation data for co-training, so that the performance evaluated in simulation can effectively reflect the performance in the real world.

**Cross-Embodiment Co-Training.** A large amount of work (Doshi et al., 2024; Yang et al., 2024; O'Neill et al., 2024; Physical Intelligence, 2025; Yuan et al., 2025) has explored using cross-embodiment robot data for co-training, where a single policy is trained with multiple embodiments with a unified architecture and action representation. The main domain gaps include the visual appearance and the embodiment-physics gap. Human data is a special case of them, which can be directly treated as another embodiment. These works show that diverse robot pretraining can produce transferable internal representations (Kareer et al., 2025b). Some works (Cai et al., 2025; Kareer et al., 2025a; Punamiya et al., 2025) utilize representation alignment regularization, such as optimal transport and adversarial discriminator. Another line of work (Xu et al., 2025; Lepert et al., 2025b;a) forces input data distribution alignment via image editing.

**Non-Robot Data Co-Training.** Internet-scale multimodal data without action labels is another valuable co-training resource. Numerous works (Physical Intelligence, 2025; Lee et al., 2025; Zawalski et al., 2024; Zitkovich et al., 2023; Lin et al., 2026) have adopted VL datasets, which contain rich commonsense knowledge, planning and spatial information for co-training in VLAs. These works have demonstrated that co-training with VLM data can transfer knowledge from other modalities. Also, a number of works try to utilize videos for policy co-training. Some works (Kareer et al., 2025a; Lepert et al., 2025a; Luo et al., 2025) explicitly extract action labels but with the problem of accuracy; some works (Ye et al., 2024; Zhou et al., 2024) explored latent action representations by encoding changes between video frames; other works (Li et al., 2025; Zhu et al., 2025; Liang et al., 2025) propose to co-train the action- and actionless video data within a unified architecture.

## A.2. Representation Learning in Domain Adaptation

Co-training can also be viewed as semi-supervised or unsupervised domain adaptation. A central theme in domain adaptation is learning representations that enable knowledge transfer across domains while mitigating distribution shift. Early theoretical work formalized this goal by relating target error to source error and representation-level domain discrepancy, motivating the pursuit of domain-invariant features through shared embeddings or feature transformations (Ben-David et al., 2010; Mansour et al., 2009). Building on this foundation, many practical approaches explicitly align representations across domains, including discrepancy-based methods that minimize statistical distances such as maximum mean discrepancy (MMD) (Long et al., 2015), OT-based alignment (Courty et al., 2017; Damodaran et al., 2018), and adversarial domain adaptation methods that encourage indistinguishability (Ganin et al., 2016; Tzeng et al., 2017).

# B. Theoretical Details

## B.1. Demonstration of Empirical Optimal Score Function in Co-Training

Diffusion models define a forward corruption process that maps data samples to a simple reference distribution over timesteps $t \in (0, 1)$. In the case of Gaussian perturbations, this process can be written as

$$x^t = \alpha_t x^0 + \sigma_t \epsilon \,,$$

where $\epsilon \sim \mathcal{N}(\mathbf{0}, \mathbf{I}_d)$. Models are trained to learn the underlying score field of this process, which is then used at inference to generate samples via reverse-time denoising.

**Learning Objective.** Given training data $\mathcal{D} = \{x_i\}_{i=1}^N$, under a Gaussian probability path, the learning objective can be written in the denoising score matching form:

$$\mathcal{L}_{origin}(t) := \mathbb{E}_{x_i \sim \mathcal{D}, \epsilon \in \mathcal{N}(\mathbf{0}, \mathbf{I}_d)}[||\epsilon - \epsilon_\theta(x^t, t)||_2^2] . \tag{10}$$

There exists an analytical optimal solution for Eq. (10):

$$s^*(x^t, t) = -\epsilon^*(x^t, t)/\sigma_t = \frac{1}{\sigma_t^2}[\alpha_t \mathbb{E}[x^0|x^t] - x^t] . \tag{11}$$

And we have

$$\mathbb{E}[x^0|x^t] = \frac{\sum_{i=1}^N p(x^t|x^0 = x_i) \cdot x_i}{\sum_{i=j}^N p(x^t|x^0 = x_j)} \tag{12}$$

where $p(x^t|x_i^0) = \mathcal{N}(x^t; \alpha_t x_i, \sigma_t^2 \mathbf{I_d})$.

Then suppose we have source and target datasets as $\mathcal{D}_T = \{x_i\}_{i=1}^N$ and $\mathcal{D}_S = \{x_j\}_{j=1}^M$, co-train a diffusion model with mixing ratio $w$, this gives us the training objective as:

$$\mathcal{L}_w(t) := w \cdot \mathcal{L}_{\mathcal{D}_T} + (1 - w) \cdot \mathcal{L}_{\mathcal{D}_S} \tag{13}$$

Similarly, we can get the analytical optimal score function as:

$$s_w^*(x^t, t) = \hat{w}_t \cdot s_t^*(x^t, t) + \hat{w}_s \cdot s_s^*(x^t, t) \tag{14}$$

where

$$\hat{w}_t := \frac{w p_t(x^t)}{w p_t(x^t) + (1 - w) p_s(x^t)} \tag{15}$$

*Proof.* To find $f(B)$ that minimizes $L(f) = w \mathbb{E}_t[\|A - f(B)\|^2] + (1-w)\mathbb{E}_s[\|A - f(B)\|^2]$, we define a mixture probability density $p_w(a, b) = w p_t(a, b) + (1 - w) p_s(a, b)$.

By expressing the expectations as integrals, we have:

$$L(f) = \iint \|a - f(b)\|^2 \left( w p_t(a, b) + (1 - w) p_s(a, b) \right) da\, db$$

$$= \iint \|a - f(b)\|^2 p_w(a, b)\, da\, db = \mathbb{E}_w[\|A - f(B)\|^2].$$

It is a standard property of Hilbert spaces of random variables that the MSE is minimized by the conditional expectation $\mathbb{E}_w[A \mid B]$:

$$f^*(b) = \int a\, p_w(a \mid b)\, da = \frac{\int a\, p_w(a, b)\, da}{p_w(b)} = \frac{\int a\, (w p_t(a, b) + (1 - w) p_s(a, b))\, da}{w p_t(b) + (1 - w) p_s(b)}.$$

As $\int a\, p_i(a, b)\, da = \mathbb{E}_i[A \mid B = b] p_i(b)$ for $i \in \{t, s\}$, we obtain the optimal solution:

$$f^*(B) = \frac{w p_t(B) \mathbb{E}_t[A \mid B] + (1 - w) p_s(B) \mathbb{E}_s[A \mid B]}{w p_t(B) + (1 - w) p_s(B)}.$$

This shows that the co-trained optimal predictor is a weighted combination of the optimal predictors in each domain. □

### B.2. Derivation from Gaussian Distribution to Softmax

Starting from Eq. 14 proved above:

$$s_w^*(a^t, t) = \hat{w}_t \cdot s_t^*(a^t, t) + \hat{w}_s \cdot s_s^*(a^t, t) \tag{16}$$

where the mixing weight $\hat{w}_t$ is given by:

$$\hat{w}_t := \frac{wp_t(a^t)}{wp_t(a^t) + (1-w)p_s(a^t)} \tag{17}$$

Combining Eq. 12 and expanding $p_t(a^t)$, we can have:

$$s_w^*(a^t, t) = \sum_{k=1}^{N+M} \frac{w_k p(a^t|a_k)}{\sum_j w_j p(a^t|a_j)} \cdot \left(\frac{\alpha_t a_k - a^t}{\sigma_t^2}\right) = \sum_{k=1}^{N+M} \frac{w_k p(a^t|a_k)}{\sum_j w_j p(a^t|a_j)} \cdot s_k^*(a^t, t) \tag{18}$$

We recognize the fractional term as the posterior weight $g_k$, which can be computed via the Softmax function using the distance metric $r_k = \frac{||a^t - \alpha_t a_k||}{\sigma_t \sqrt{d}}$:

$$g_k = \texttt{Softmax}(\ln(w_k) - r_k^2 * d/2) \tag{19}$$

By further simplification, we obtain the final simplified form:

$$\begin{aligned}
s_w^*(a^t, t) &= \frac{\alpha_t}{\sigma_t^2} \cdot \left(\sum_{i_t}^{N} g_{i_t} a_{i_t} + \sum_{i_s}^{M} g_{i_s} a_{i_s}\right) - \frac{a^t}{\sigma_t^2} \\
&= \sum_{i_t}^{N} g_{i_t} s_{i_t}^* + \sum_{i_s}^{M} g_{i_s} s_{i_s}^*
\end{aligned} \tag{20}$$

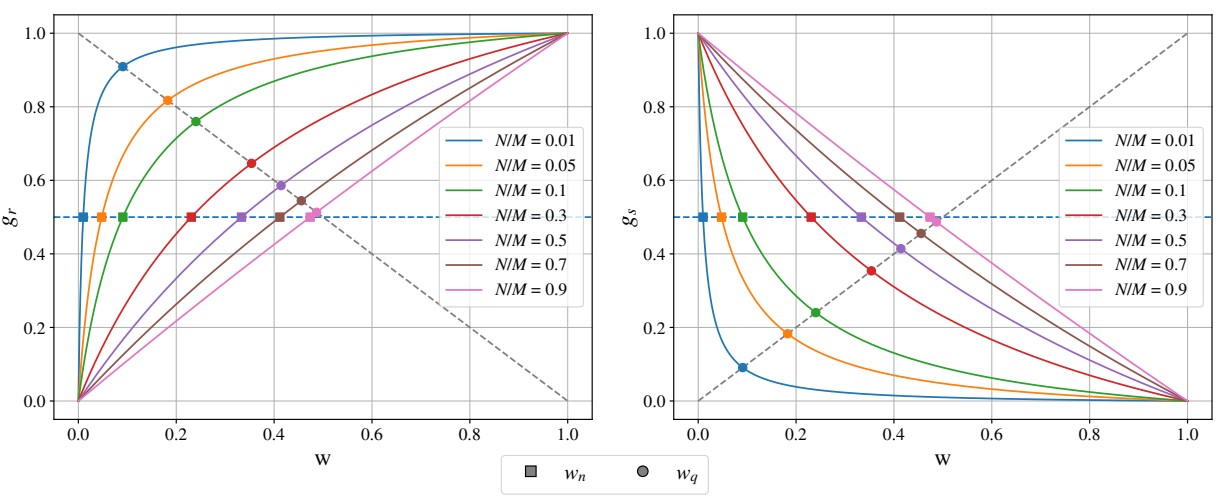

*Figure 7.* N/M scaling

### B.3. Empirical Estimation

**[Fact 1. Gaussian concentration in high-dimensional space]:** for a standard Gaussian vector $\epsilon \sim \mathcal{N}(\mathbf{0}, \mathbf{I}_d)$, we have $||\epsilon|| = \sqrt{d}(1 + \mathcal{O}(1))$ with high probability. So for each component in $p_r(x_t), p_s(x_t)$, it concentrates on a set of thin spherical shells centered at $\alpha_t x_i$ with radius $\sigma_t \sqrt{d}$.

**[Empirical evidence 1. Data points separation]:** for most of the timesteps $t$ that are not too large, in high-dimensional data space, these shells are non-overlapping as $\sigma_t \sqrt{d} \ll \min_{i,j} ||\alpha_t x_i - \alpha_t x_j||$. We show this in the action-chunking space of diffusion policy as shown in Fig. 8.

These make the softmax weights extremely imbalanced, which is nearly biased towards the nearest training data.

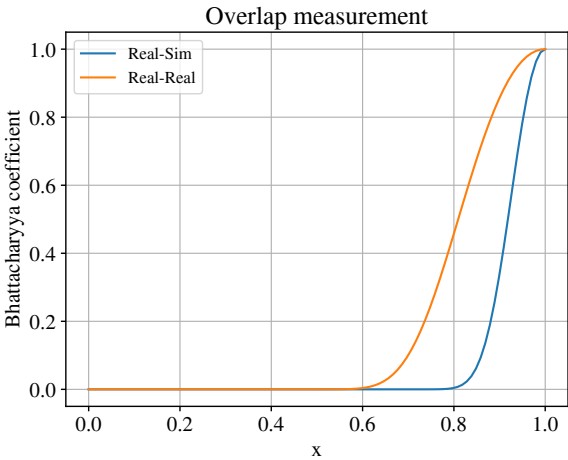

*Figure 8.* Empirical measurement about training sample overlapping using Bhattacharyya coefficient. We compute it in the same way as in Song et al. (2025), but include the distances between observation features as we are modeling the conditional distribution.

**[A Special Case Study]**: Let's assume an ideal case where the action chunks are uniformly distributed across different states on the trajectories, so the ratio of the number of trajectories between simulation and real is similar to the ratio of the number of action chunks of different states between simulation and real. A special case is that the nearest data points in $\mathcal{D}_r$ are similarly close to the nearest data points in $\mathcal{D}_s$, then we can have:

$$s_w^*(a^t, t) \approx \frac{\alpha_t}{\sigma_t^2} \left[ \frac{w/N \cdot \exp(-r_r^2/2)}{w/N \cdot \exp(-r_r^2/2) + (1-w)/M \cdot \exp(-r_s^2/2)} \cdot x_r + \frac{(1-w)/M \cdot \exp(-r_s^2/2)}{w/N \cdot \exp(-r_r^2/2) + (1-w)/M \cdot \exp(-r_s^2)} \cdot x_s \right] \tag{21}$$

Then we have $\frac{g_r}{g_s} = \frac{1-w_N}{w_N} \cdot \frac{w}{1-w} \cdot \exp(\frac{r_s^2 - r_r^2}{2}) = \frac{1-w_N}{w_N} \cdot \frac{w}{1-w} \cdot \exp(\frac{|a_t - (a_r + a_s)/2| \cdot |a_r - a_s|}{2\sigma_t^2 d})$, also we can draw the relation between $g_r, g_s$ and $w$. We can find that as the ratio of sim/real increases, the choice of $w$ should be more robust, which aligns with prior observations from Wei et al. (2025). And $g_r(w_n) = g_s(w_n) = 1/2$, which aligns with our intuition.

Since we find that the curve will be extremely steep for small $N/M$, we posit that the balanced mixing ratio should make good use of both real and sim, but place more weight on real. Take the intersection point between $g_r, g_s$ with diagonal $g = 1 - w$ and $g = w$, the coordinate of intersection point in $g_r$ is $(w_q, 1 - w_q)$ where $w_q = \frac{\sqrt{q}}{\sqrt{q}+1}$ and $q = \frac{N}{M}$. So the relative domain weight $\frac{g_r}{g_s}$ should be between $(1, \sqrt{\frac{M}{N}})$, and balanced mixing ratio should be between $(w_n, w_q \approx \sqrt{\frac{N}{M}})$ when $N \gg M$. With this principle, this range is around $(0.016, 0.13)$ in our experiments which also aligns with our experimental observations.

## C. Training Details

The original images are captured by the camera at a resolution of 720×1280. During preprocessing, the images are downsampled to 90×106, followed by random cropping to 84×84 during training and center cropping during testing. The policy takes stacked history images and robot proprioceptive inputs (joint and gripper positions) as input, and outputs 7-DOF target joint positions along with the gripper action.

The overall training procedure of *CFG-ADDA* is summarized in Algo. 1. We implement the discriminator as a simple 3-layer MLP. Across all of our experiments, we use a batch size of 256, dropping probability $p = 0.2$ and a weighting coefficient $\lambda = 0.1$. To ensure the discriminator can learn in a compact feature space and also provide effective reverse gradients to the policy, we only start to train the discriminator and compute the discriminative loss after a warming step of 5000.

The implementation of *CFG* and *ADDA* can be seen as only keeping the corresponding computation part in Algo. 1.

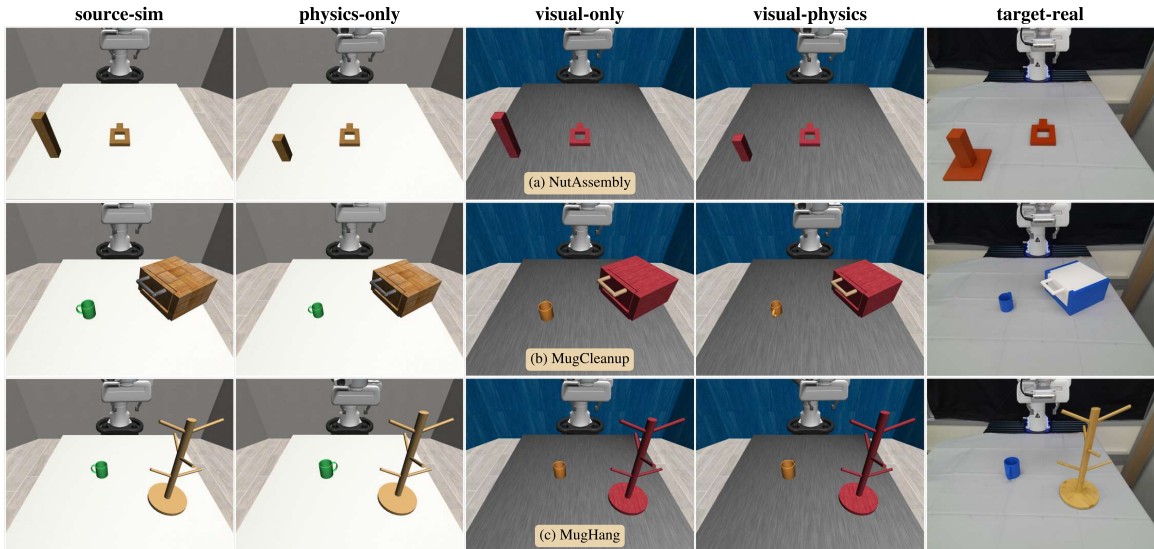

*Figure 9.* **Visualizations of the designed sim-and-sim and sim-and-real tasks.** In the *physics-only* setting, we vary the physical parameters of objects, including mass, friction, and size, while keeping the appearance similar. Across all target environments, we tune the robot controller configurations to match those in the real world.

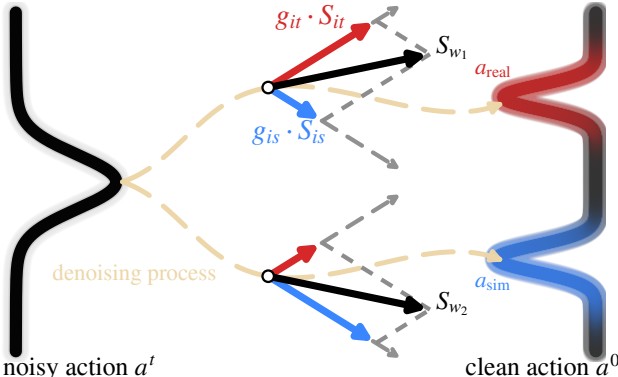

*Figure 10.* Importance reweighting reshapes the learned action distribution via reweighting score functions during training time.

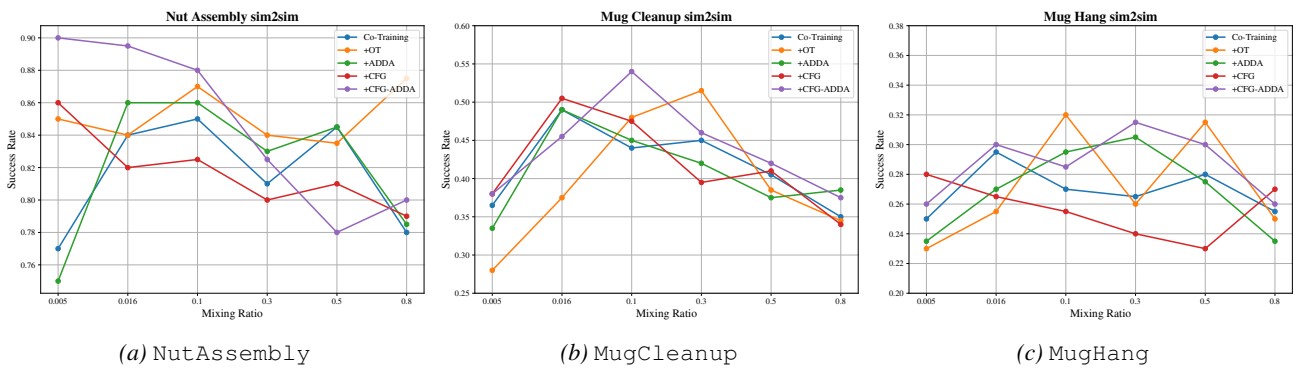

*(a)* NutAssembly      *(b)* MugCleanup      *(c)* MugHang

*Figure 11.* **Detailed results of sim-and-sim evaluations.**

---

**Algorithm 1** : CFG-ADDA

---

1: **Input:** Source dataset $\mathcal{D}_s$, target dataset $\mathcal{D}_t$, mixing ratio $w$, randomly initialized $f_\phi, \pi_\theta$ and discriminator $Disc_\mu$.
2: **for** iterations $t = 1$ to T **do**
3:      Sample data $\{(o_{i_t}, a_{i_t})\}$ with size $N \cdot w$ from $\mathcal{D}_t$ and $\{(o_{i_s}, a_{i_s})\}$ with size $N \cdot (1-w)$ from $\mathcal{D}_s$
4:      Create one-hot embeddings as environment labels $\{c\} = \{c_{i_t}\} \cup \{c_{i_s}\}$
5:      Randomly set environment labels as $\varnothing$ with probability $p$
6:      Compute features $z_i = f_\phi(o_i)$
7:      Compute discriminator loss $\mathcal{L}_{disc} = -\mathbb{E}_{z_i}[\log Disc_\mu(z_{i_s})] - \mathbb{E}_{z_i}[\log(1 - Disc_\mu(z_{i_t}))]$
8:      Simply concatenate $z_i$ and $c_i$, compute behavior cloning loss $\mathcal{L}_w(z_i, c_i, a_i)$
9:      Update $f_\phi, \pi_\theta, Disc_\mu$ with gradient of $\mathcal{L}_w(\phi, \theta) + \lambda \cdot \mathcal{L}_{disc}(\mu)$
10: **end for**

---

# D. Additional Experiments Results

## D.1. Detailed Results of Sim-and-sim Evaluation

We provide the detailed sim-and-sim evaluation results of comparing different co-training techniques. In Fig. 6, we categorize $0.016, 0.1, 0.3$ as balanced mixing ratio while the others as unbalanced mixing ratio. Compared to the previous techniques which only emphasizes one side of structured representation alignment, our simple combination brings more stable improvement across different mixing ratios.

## D.2. Details on Representation Alignment

**Details of Measurement.** We use UMAP dimension reduction for qualitative visualizations, and measure the Gromov-Wasserstein distance (Mémoli, 2011) and the Wasserstein distance (Rüschendorf, 1985) to show local geometric similarity and global distributional distance for quantitative comparison. We randomly sample a large batch (1024) of data from each domain and normalize the features within each domain independently before quantitative measurement. We assume the down-sampled distribution can represent the distribution of the complete dataset. To compare the Gromov-Wasserstein distance across multiple policies, we normalize the cost matrix by the pooled mean of each domain, thereby controlling for differences in value amplitudes.

**More Feature Visualizations.** As shown in Fig. 17, the changing trends of the two distances with mixing ratios are similar across different tasks. When $w$ is in the range of balanced mixing ratio (roughly between $w_n$ and $0.5$), the representations of simulation and real-world observations share more similar geometric structures and exhibit greater overlap.

## D.3. Robustness to Different Policy Architectures

As our main experiments are all conducted on encoder-decoder transformer-based diffusion policy, to further support the robustness of our findings to different architectures, we do another series of experiments with the same setting on two other architectures — decoder-only transformer-based model and CNN-based U-Net diffusion policy. The results in Fig. 12 show that the best performance consistently lies in a narrow range ($[0.016, 0.3]$), indicating robustness beyond the architecture used in the main paper.

## D.4. Comparison to "Simulation Pre-training+Real Fine-tine"

Although some prior work shows that it is generally less effective than co-training, as this still remains as an important baseline, we include a direct comparison here. We pre-train with simulation data only for $\sim 130k$ steps and fine-tune with real data for $\sim 120k$ steps. We evaluate all checkpoints along the training and report the highest success rate. Our result is consistent with prior findings.

## D.5. A Formal Statement of Mixing Ratio Selection Guideline

The main experiments in the paper rely a heuristic defined mixing ratio grids. Based on the above analysis, we provide a guideline for selecting mixing ratios for co-training, which we hope will be useful to the community. We conduct some additional experiments to further support the effectiveness of our guideline. We co-train with 3 sets of different dataset sizes

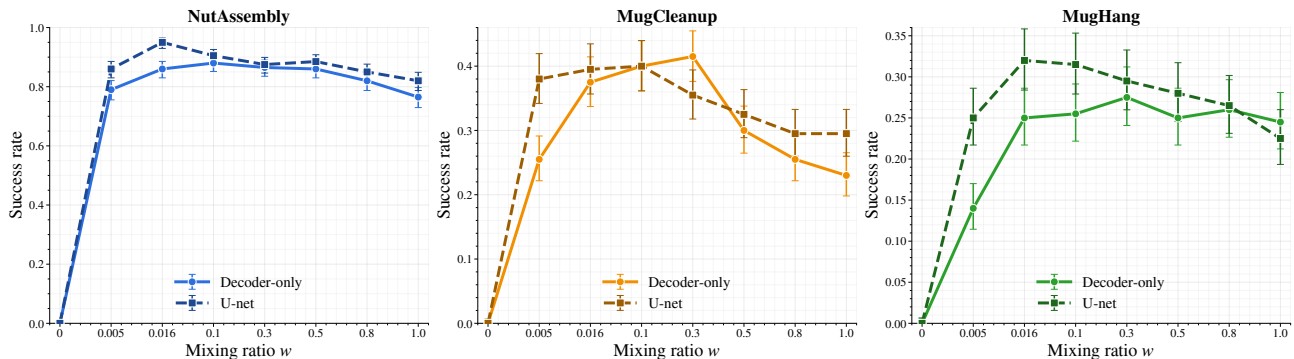

Figure 12. **Balanced mixing ratios are robust to different policy architectures.** Each data point is computed over three policy checkpoints, and each checkpoint is evaluated for 200 trials. The best performance is consistently achieved in the range of $(0.016, 0.3)$.

| Task | Pretrain+FT | Co-training |
|------|-------------|-------------|
| NutAssembly | 0.77 | **0.925** |
| MugCleanup | 0.215 | **0.495** |
| MugHang | 0.225 | **0.26** |

Table 2. Performance comparison with Pre-train+FT.

– 10:3000, 50:500, and 50:100 – and sweep the mixing ratios.

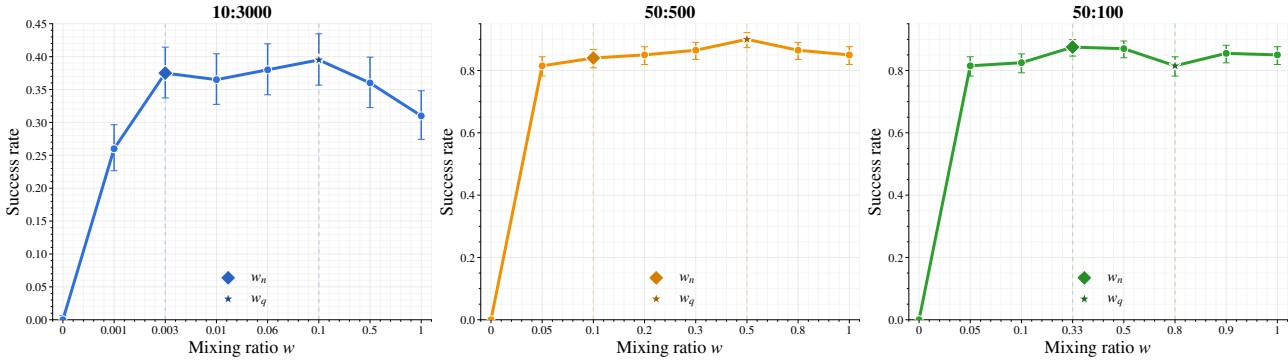

Figure 13. **Experiments on different dataset sizes.** Each data point is computed over three policy checkpoints, and each checkpoint is evaluated for 200 trials. The best performance is consistently achieved in the range of $(w_n, w_q)$, which supports our guideline below.

### D.6. Three Regimes of "Representation" in Co-Training

In Sec. 2.1 of the main paper, we point out three different hypothetical scenarios of representation in co-training – overlapping, structured aligned, and disjoint. As mentioned above, we use two coupled and measurable properties to characterize the concept of "structured representation alignment". So it can be mathematically defined as follows:

$$\mathcal{SRA}(p_s, p_t) = (\mathcal{M}_{align}, \mathcal{D}_{disc})$$

where $\mathcal{M}_{align} = \mathcal{W}(p_s(z), p_t(z))$ and $\mathcal{D}_{disc} = \frac{1}{2}\sum_{k\in\{s,t\}} \mathbb{E}_{z\sim p_k(z)}\left[\max_{a^t} p_k(a^t \mid z)\right]$. We design another set of experiments to show the existence of these three regimes: We introduce an additional and complementary control knob via discriminator regularization, which directly modulates domain discernibility. Specifically, we train models with discriminator loss weights of $\{0, 0.05, 0.5\}$, and within each setting sweep the mixing ratio to vary alignment. This allows us to populate a substantially broader region of the (alignment, discernibility) space.

Empirically, these settings correspond to different regimes: (1) No discriminator (0): representations remain weakly aligned (high WD) but highly distinguishable (high discernibility), corresponding to the disjoint regime (and partially covering the boundary toward structured alignment). (2) Moderate discriminator (0.05): representations become better aligned

---

**Algorithm 2** Guideline for Co-Training Mixing Ratio Selection

---

**Require:** Source dataset size $N$, target dataset size $M$ with $M > N$
**Require:** Optional desired target contribution $q$ (e.g., $q = 0.8$)
**Ensure:** A narrowed search range $(w_n, w_q)$ for the mixing ratio
 1: Compute the natural mixing ratio

$$w_n = \frac{N}{N + M}.$$

 2: Use $w_n$ as the lower bound of the search range.
 3: **if** $M/N > 5$ **then**
 4:    Set the upper bound as

$$w_q = \sqrt{\frac{N}{M}}.$$

 5: **else**
 6:    Set a desired target contribution ratio $q$ (e.g., $q = 0.8$).
 7:    Compute the upper bound as

$$w_q = \frac{N * q}{(1 - q) * M + N * q}.$$

 8:    Optionally, set the upper bound empirically to $0.5$, which is empirically enough.
 9: **end if**
10: Adjust $w_n, w_q$ upward if the source-target domain gap is large.
11: Consider domain gaps from visual appearance, physics, and embodiment.
12: Since no formal estimator is assumed, apply this adjustment accordingly.
13: Perform a simple search for the final mixing ratio within

$$(w_n,\ w_q).$$

---

(lower WD) while preserving domain-specific structure (high discernibility), corresponding to the desired structured aligned regime. (3) Strong discriminator (0.5): representations are strongly aligned (low WD), but domain-specific information is suppressed (low discernibility), corresponding to the overlapping regime. Within each regime, varying the mixing ratio further modulates the degree of alignment, producing a set of points spanning different Wasserstein distances.

We provide the corresponding 2D visualization (including Wasserstein distance) in Fig. 14. We observe: - In the overlapping regime, the correlation between performance and WD is reversed, consistent with observations in physics-only settings. - In the disjoint regime, the correlation follows the standard co-training trend (better alignment improves performance). - In the structured aligned regime, the correlation remains positive but weaker, indicating a distinct intermediate behavior.

Taken together, these regimes exhibit a non-monotonic relationship, resulting in a characteristic **inverted-U (or U-shaped) curve** when alignment and discernibility are jointly considered. This unifies the three regimes as different regions of a single underlying mechanism.

| | NutAssembly | MugCleanup | MugHang | Avg |
|---|---|---|---|---|
| co-training | 0.85 | 0.44 | 0.27 | 0.52 |
| +discrimination | 0.79 | 0.415 | 0.215 | 0.47 |

*Table 3.* **Performance degrades after adding penalty for representation alignment.** The experiments are run with $w = 0.1$.

### D.7. Ablations on Guidance Scale.

In contrast to using only the positive values, we sweep $\lambda$ in the interval of $(-2, 2)$ with both *CFG* and *CFG-ADDA*. As shown in Fig. 16, our proposed method consistently outperforms *CFG* with different guidance scales. Besides, both methods exhibit improvement with $\lambda = -0.5$. So instead of setting $\lambda > 0$ to amplify the action gaps in a traditional way, we advocate

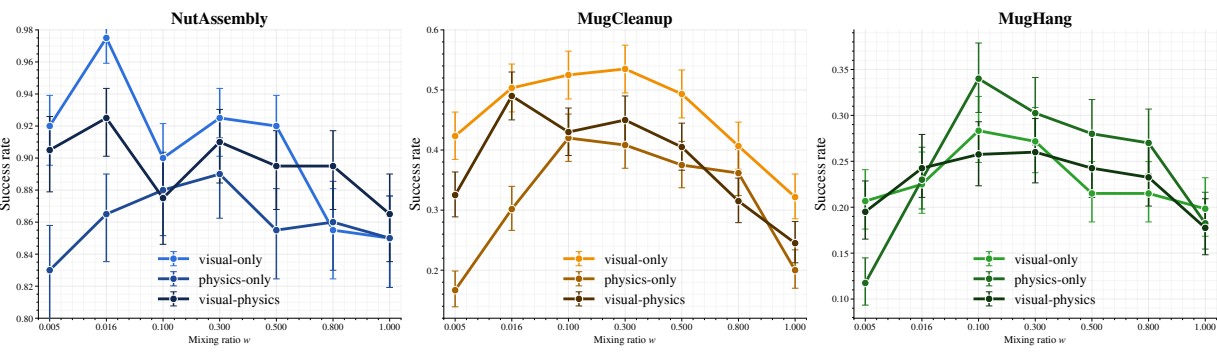

*(a)* **Three regimes exhibit U-shape correlation pattern.**

*(b)* **Three regimes on two-axises – representation alignment and domain discernibility.**

*Figure 14.* **Existence of three regimes in co-training.**

*Figure 15.* **Success rate of sim-and-sim co-training with different domain gaps.** Each data point is computed over three policy checkpoints, and each checkpoint is evaluated for 200 trials. The best performance is consistently achieved in the range of $(0.016, 0.3)$.

setting $\lambda < 0$ to actively transfer knowledge from the surrogate domains during inference.

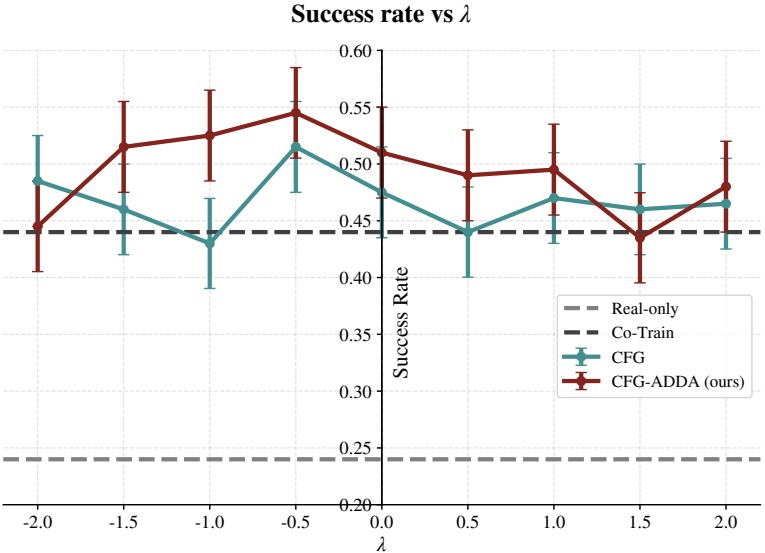

*Figure 16.* **Performance of different co-training methods on *MugCleanup* with $w = 0.1$ in sim-and-sim settings.** The plots are similar for all tasks and balanced mixing ratios.

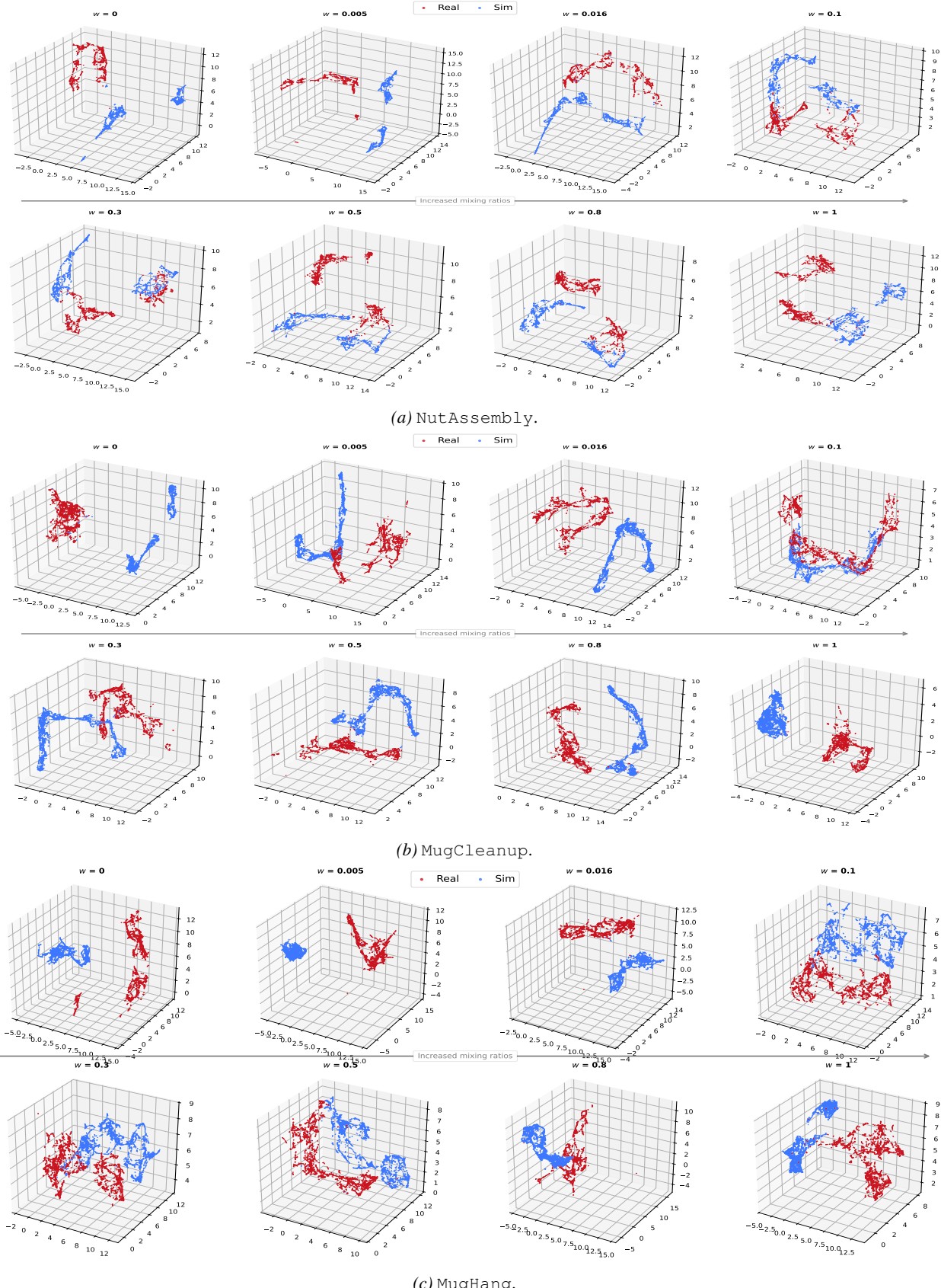

*(a)* NutAssembly.

*(b)* MugCleanup.

*(c)* MugHang.

*Figure 17.* **More UMAP visualizations of the observation representations.**

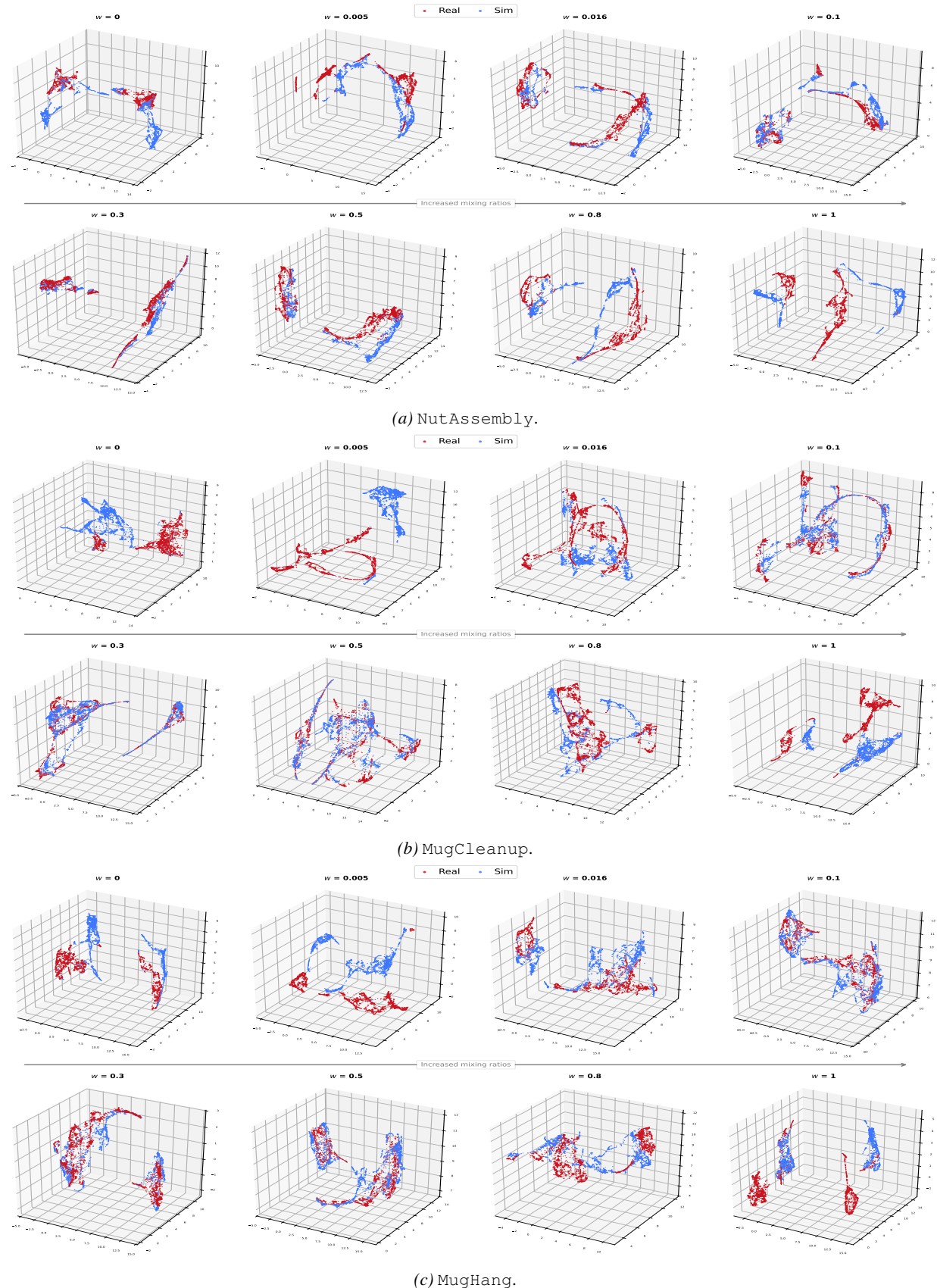

*(a)* `NutAssembly`.

*(b)* `MugCleanup`.

*(c)* `MugHang`.

*Figure 18.* **More UMAP visualizations of the deeper layer representations.**

