# OpenReview forum: "A Mechanistic Analysis of Sim-and-Real Co-Training in Generative Robot Policies"
_ICML.cc/2026/Conference — ICML 2026 regular_

### Official Review · Reviewer_AFgA · 2026-03-12

**Soundness:** 2
**Presentation:** 2
**Significance:** 2
**Originality:** 3
**Overall Recommendation:** 4
**Confidence:** 3

**Summary:**

This paper studies co-training on sim and real data for robotic policy learning. Two effects are analyzed: 1) the effect of the ratio of training samples from the two domains, and 2) the "structured representation alignment" of embeddings between the two domains. It is argued that #1 affects #2, and #2 is the causal factor in determining robot performance. The paper also contributes a new co-training algorithm that seeks to achieve better representation alignment. This pays off in a modest improvement in task performance.

**Compliance With Llm Reviewing Policy:**

Affirmed.

**Final Justification:**

The rebuttal clarifies many issues and charts a path toward significant improvement of the paper, particularly in terms of a better formalization of structured representation alignment. The additional experiments showing the three regimes is also a very nice addition. Therefore I advocate accepting.

**Key Questions For Authors:**

1. Can the authors provide a precise definition of structured representation alignment, e.g., an equation that measures it?
2. Can the authors respond to weakness #3, especially explaining why we don't see U-shaped curves in Figure 5, and how those graphs validate, or do not, the three regimes mentioned in the theoretical analysis (disjoint, aligned, overlapping)?

**Limitations:**

Yes.

**Strengths And Weaknesses:**

Strengths:
1. The mixing ratio seems like a critical parameter to understand, and the paper provides convincing evidence that it has a sweet spot.
2. Structured representation alignment is an interesting concept, and the theoretical argument for its importance is convincing (although the precise meaning is vague).
3. CFG-ADDA seems reasonable and gives a boost in performance.

Weaknesses:
1. There’s no precise definition of structured representation alignment. The paper would be clearer if it were defined mathematically, and measurably. Then the experiments could systematically vary this measure and see how it affects performance. As is, the experiments instead vary other measures (such as Wasserstein distance), which are related to alignment, but perhaps not direct measures of it. Alternatively, the paper could be framed in terms of analyzing one of these concrete measures, e.g., change the intro to be "We identify two key factors, i.e., balanced data mixing ratio $w$ and Wasserstein distance..."
2. The toy experiment is hard to understand. Part of this is due to writing / figure clarity, e.g., Figure 1 uses the same symbol for both "Source Samples" and "Predictions." Beyond the clarity issues, however, I'm not sure what to take from this experiment. It's not quantitative, and it's not obvious to me that the Structured Aligned setting is qualitatively different or better than the other settings, from examination of Figure 1.
3. The experiments in Section 4 do not convince me that the “overlapping” case is actually observed, nor that it causes poor performance. Instead it seems like better alignment is beneficial in almost all the tested settings. There’s an anti-correlation in the “physics-only” setting in Figure 5, but the text only gives a vague explanation that “As we largely change the object’s physical parameters while keeping its visual appearance similar, it is harder for co-trained policies to distinguish between the two environments.” This needs better quantification. Further, if Normalized WD were directly measuring structured representational alignment, and if there were three regimes (disjoint, aligned, overlapping), and if aligned is the only good regime, then I would expect to see U-shaped curves in Figure 5, but I do not, at least not clearly enough. This undermines the whole thesis of the paper.

Minor:
1. “p-values < 0.4” —> “p-values < 0.04”

---

> ### Author Rebuttal · Authors · 2026-03-31
>
> We thank the reviewer for finding "structured representation alignment" **interesting** and providing constructive feedback! Please see our responses addressing the specific concerns below:
>
> >W1: There’s no precise definition of structured representation alignment. The paper would be clearer if it were defined mathematically, and measurably.Then the experiments could systematically vary this measure and see how it affects performance.
>
> Thanks for this insightful feedback. We agree that formalizing the concept of "structured representation alignment" strengthens the theoretical foundation of our paper. Starting from the Eq.3 in the paper, let $p_k(a^t, z)$ denote the joint distribution of the action $a^t$ and representation $z = f_\phi(o)$ for domain $k \in \{s, t\}$, we have:
>
> $$
> p\_k(a^t, z) = \frac{1}{|\mathcal{D}\_k|} \sum\_{i\sim \mathcal{D}\_k} p(a^t|z\_i)\cdot K(z, z\_i)
> $$
>
> We can formally define the measure of Structured Representation Alignment as the discrepancy between these joint distributions, e.g., $\mathcal{SRA}(p\_s(a^t, z), p\_t(a^t, z))$. Our insight is that it is not a scalar metric, but a two-coordinate metric:
>
> $$\mathcal{SRA}(p\_s,p\_t) = (\mathcal{M}\_{align}, \mathcal{D}\_{disc})$$
>
> where $\mathcal{M}\_{align}=\mathcal{W}(p\_s(z),p\_t(z))$ and $\mathcal{D}\_{\mathrm{disc}} = \frac{1}{2} \sum\_{k \in \{s,t\}} \mathbb{E}\_{z \sim p\_k(z)} \left[ \max\_{a^t} p\_k(a^t \mid z) \right]$.
>
> So two measurable objectives would be: (1)Alignment on marginal distribution: Minimizing the distance between the marginal representation distributions $p_s(z)$ and $p_t(z)$, which we explicitly measure using the Wasserstein distance. (2)Discernibility on conditional distribution: Preserving the separability of the conditional distributions $p_k(a^t|z)$, which ensures representations do not collapse and actions remain discernible. We explicitly measure this via the performance of a trained classifier, and use domain label $p(l|z)$ as a proxy. This is a disentanged definition, while we believe a single unified definition might be a potential future direction.
>
> > W2: The toy experiment is hard to understand. Part of this is due to writing / figure clarity, e.g., Figure 1 uses the same symbol for both "Source Samples" and "Predictions." Beyond the clarity issues, however, I'm not sure what to take from this experiment. It's not quantitative, and it's not obvious to me that the Structured Aligned setting is qualitatively different or better than the other settings, from examination of Figure 1.
>
> Thanks for the reminder! The symbols are too small, actually the "source samples" are blue while "predictions" are green. Besides, we provide a quantitative measurement on the prediction error [here](https://https://dynamics-humanoid.github.io/cotraining-anonymous/) since we know the ground truth mapping, which also justifies that structured alignment is the main driver. In figure.1, the good predictions should be closer to the red manifold, so in certain setting (w=0.1 when structured aligned), the predictions can even OOD generalize. We provide [videos](https://https://dynamics-humanoid.github.io/cotraining-anonymous/) for better comparison. We will fix all of these in the next version of the paper.
>
> > W3: Can the authors respond to weakness #3, especially explaining why we don't see U-shaped curves in Figure 5, and how those graphs validate, or do not, the three regimes mentioned in the theoretical analysis (disjoint, aligned, overlapping)?
>
> - Firstly, we also train the classifier on the physics-only features. The accuracy drops to ~90%, while achieves ~100% consistently in other settings. So the model has relatively low discerniblity in this setting, and we observe negative correlation between alignment and performance. That's one reason why we define the structured representation alignment with two properties.
> - Normalized WD only measures one dimension of structured alignment, but relates to the discerniblity. In the Figure.5, the only control variable is the mixing ratio. We argue that changing mixing ratio only cannot cover three regimes at the same time, but at least two. For example, in visual-only, visual-physics and target-real, changing the mixing ratio shapes the representation between structured aligned and disjoint regime; while in physics-only, as lacking the discernibility, the representation is shifted between overlap and structured aligned regime. So we cannnot see a U-shaped curves.
> - Actually the WD in physics-only (10.88, without normalization) is smaller than the other settings (11.68, 11.81 and 12.72), so if we draw the curves across the setting, we can see the U curve. But that's not a rigorous evidence. More straight-forward way to see the effect of overlap is in the controlled toy example.

---

> > ### Author Rebuttal · Reviewer_AFgA · 2026-04-03
> >
> > Thanks for the rebuttal!
> >
> > I like these alignment and discernability measures. I think this will clarify the paper a lot.
> >
> > I think W3 remains a weakness, as the paper still does not clearly show the three regimes. Perhaps you can plot the data as a heatmap, with the x-axis as alignment (WD) and the y-axis as discernability, and the color as performance? Then it should be the case that only when you have high alignment and high discernability do you have high performance, right? I guess there should be three regions in this map: 1) high WD, high discernability --> disjoint regime, 2)  low WD, high discernability --> structure aligned regime, 3)  low WD, low discernability --> overlapping regime?
> >
> > Note that this link seems to be down: https://https//dynamics-humanoid.github.io/cotraining-anonymous/

---

> > > ### Author Response · Authors · 2026-04-07
> > >
> > > Thanks for your recognition, and for this valuable suggestion! We agree that visualizing performance in the joint space of alignment and discernability provides a clearer and more intuitive illustration of the three regimes. We also apologize for the delay—since results from different settings cannot be directly combined, we conducted massive extra experiments to properly construct this analysis.
> > >
> > > As we noted earlier, a key challenge is that varying the mixing ratio alone provides only a single control variable, while alignment (measured by Wasserstein distance) and discernability (measured by domain classification accuracy) are **two coupled properties**. Consequently, a mixing-ratio sweep typically traces only a 1D trajectory in this 2D space.
> > >
> > > To address this, we introduce an additional and complementary control knob via **discriminator regularization**, which directly modulates domain discernability. Specifically, we train models with discriminator loss weights of \{0, 0.05, 0.5\}, and within each setting sweep the mixing ratio to vary alignment. This allows us to populate a substantially broader region of the (alignment, discernability) space.
> > >
> > > Empirically, these settings correspond to different regimes:
> > > - **No discriminator (0)**: representations remain weakly aligned (high WD) but highly distinguishable (high discernability), corresponding to the *disjoint regime* (and partially covering the boundary toward structured alignment).
> > > - **Moderate discriminator (0.05)**: representations become better aligned (lower WD) while preserving domain-specific structure (high discernability), corresponding to the desired *structured aligned regime*.
> > > - **Strong discriminator (0.5)**: representations are strongly aligned (low WD), but domain-specific information is suppressed (low discernability), corresponding to the *overlapping regime*.
> > >
> > > Within each regime, varying the mixing ratio further modulates the degree of alignment, producing a set of points spanning different Wasserstein distances.
> > >
> > > A practical challenge is that classification accuracy within each regime is highly concentrated (e.g., $\sim50\%$) for strong regularization and ($\sim 98\%-99\%$) for weak/moderate regularization), making it difficult to directly visualize all settings in a single 2D plane. Therefore, we present a **2×2 layout**, where each subplot corresponds to one regime, plotting alignment (WD) vs. performance (success rate), while the outer axes provide the conceptual alignment–discernability interpretation.
> > >
> > > For reference, the success rates are summarized below:
> > >
> > > | Task         | Reg | w=0.05 | w=0.16 | w=0.1 | w=0.3 | w=0.5 | w=0.8 |
> > > |--------------|-----|--------|--------|-------|-------|-------|-------|
> > > | NutAssembly  | 0    | 0.77  | 0.84  | 0.85 | 0.81 | 0.845 | 0.78 |
> > > |              | 0.05 | 0.85  | 0.83  | 0.89 | 0.795 | 0.845 | 0.76 |
> > > |              | 0.5  | 0.685 | 0.645 | 0.245 | 0.32 | 0.1   | 0.36 |
> > > | MugCleanup   | 0    | 0.365 | 0.49  | 0.44 | 0.45 | 0.405 | 0.35 |
> > > |              | 0.05 | 0.35  | 0.505 | 0.455 | 0.44 | 0.42  | 0.33 |
> > > |              | 0.5  | 0.32  | 0.235 | 0.365 | 0.1  | 0.135 | 0.05 |
> > > | MugHang      | 0    | 0.25  | 0.295 | 0.27 | 0.265 | 0.28  | 0.255 |
> > > |              | 0.05 | 0.31  | 0.3   | 0.38 | 0.315 | 0.26  | 0.275 |
> > > |              | 0.5  | 0.17  | 0.255 | 0.215 | 0.17 | 0.185 | 0.225 |
> > >
> > > The table above shows that as alignment regularization increases, performance first improves (disjoint → structured aligned), but excessive alignment that suppresses domain-specific structure leads to performance degradation (overlapping).
> > >
> > > We provide the corresponding 2D visualization (including Wasserstein distance) **[here](https://dynamics-humanoid.github.io/cotraining-anonymous/)**. From the results (Fig.4 in the link), we observe:
> > > - In the **overlapping regime**, the correlation between performance and WD is reversed, consistent with observations in physics-only settings.
> > > - In the **disjoint regime**, the correlation follows the standard co-training trend (better alignment improves performance).
> > > - In the **structured aligned regime**, the correlation remains positive but weaker, indicating a distinct intermediate behavior.
> > >
> > > Taken together, these regimes form a **non-monotonic relationship**, resulting in a characteristic **inverted-U (or U-shaped) curve** when jointly considering alignment and discernability. This unifies the three regimes as different regions of a single underlying mechanism.
> > >
> > > Finally, we would like to clarify that the “three regimes” are a **conceptual lens** rather than the core claim of our paper. Our main contribution is to highlight the necessity of **both alignment and discernability**, supported by theoretical analysis, multi-layered empirical validation, and practical guidelines for co-training.
> > >
> > > We sincerely hope this additional analysis strengthens your assessment of our work!
> > >
> > > Besides, the correct link is: https://dynamics-humanoid.github.io/cotraining-anonymous/.

---

### Official Review · Reviewer_c6Lf · 2026-03-12

**Soundness:** 3
**Presentation:** 2
**Significance:** 3
**Originality:** 3
**Overall Recommendation:** 4
**Confidence:** 3

**Summary:**

This paper investigates why sim-and-real co-training succeeds in generative robotic policies. The authors identify that a balanced mixing ratio of data triggers structured representation alignment, allowing models to transfer simulation knowledge while maintaining the domain discernibility needed for real-world adaptation. Based on these insights, the study proposes an optimized co-training approach that outperforms existing baselines.

**Compliance With Llm Reviewing Policy:**

Affirmed.

**Final Justification:**

The rebuttal has improved my confidence in the paper’s technical contribution and positioning.

**Key Questions For Authors:**

* While ADDA aims to erase domain differences, CFG relies on domain labels. Do these two objectives conflict during joint training, and how do you ensure they don't cancel each other out?
* How do you quantitatively define the optimal "structured alignment"?
* Your findings emphasize "balanced mixing ratios." Does the theory still hold in scenarios of extreme data imbalance, such as when only 5-10 real-world trajectories are available?
* This study provides an in-depth exploration of the mechanisms underlying co-training in generative policies. However, several key studies in this field have already addressed the issues of data alignment and task composition from various dimensions. To better define the scholarly contribution of this work, it is suggested that the authors include a discussion of the following related works.

Cheng, S., Ma, L., Chen, Z., Mandlekar, A., Garrett, C.R. and Xu, D., Generalizable Domain Adaptation for Sim-and-Real Policy Co-Training. In The Thirty-ninth Annual Conference on Neural Information Processing Systems, 2025

Liu, T., Li, J., Zheng, Y., Niu, H., Lan, Y., Xu, X. and Zhan, X., Skill Expansion and Composition in Parameter Space. In The Thirteenth International Conference on Learning Representations, 2025

Pace, M.A., Dan, P., Ning, C., Bhardwaj, A., Du, A., Duan, E.W., Ma, W.C. and Kedia, K., 2025. X-Diffusion: Training Diffusion Policies on Cross-Embodiment Human Demonstrations. arXiv preprint arXiv:2511.04671.

Niu, H., Chen, Q., Liu, T., Li, J., Zhou, G., Zhang, Y., Hu, J. and Zhan, X., xted: Cross-domain adaptation via diffusion-based trajectory editing. arXiv preprint arXiv:2409.08687.

**Limitations:**

Could you discuss the framework's extensibility to transformers and propose automated methods for determining optimal data ratios ?

**Strengths And Weaknesses:**

Strengths
* Soundness: The paper provides a rigorous mechanistic analysis by combining a theoretical optimal score function derivation for diffusion policies with multi-layered empirical validation (toy examples to real-world tasks).

* Originality: It introduces the novel concept of "structured representation alignment," arguing that models must balance domain-invariance with domain-discernibility—a more nuanced perspective than traditional domain adaptation.

* Significance: By identifying balanced mixing ratios as a key driver for performance, the work offers practical, actionable insights for the growing field of sim-to-real robot learning.

* Presentation: The narrative is logical and the visualizations (e.g., latent space plots) effectively clarify complex theoretical concepts.

Weaknesses
* Theoretical Scope: The analysis is primarily centered on diffusion and flow-matching architectures; its applicability to other paradigms like autoregressive transformers remains less explored.

* Heuristic Ratio Selection: While the "balanced mixing ratio" is empirically supported, the paper lacks a formal, automated method for determining this ratio for new, diverse datasets.

* Baseline Breadth: The comparison focuses on recent robotic co-training methods but could be strengthened by including broader domain adaptation baselines from the general machine learning community.

---

> ### Author Rebuttal · Authors · 2026-03-31
>
> Thank you for your supportive and positive feedback! We try to address your specific concerns below:
>
> >W1: Could you discuss the framework's extensibility to transformers as the analysis is primarily centered on diffusion and flow-matching architectures?
>
> Thanks! This is a critical question:
> - While our analysis is developed for diffusion/flow models, the underlying mechanisms are not specific to this choice. Both diffusion and autoregressive (AR) models are trained with maximum likelihood on mixed-domain data, leading to similar learning dynamics. In particular, (i) structured representation alignment comes from sharing representations across domains under distribution shift, creating a balance between alignment and domain discernibility; and (ii) balancing effect arises from data-dependent gradients, where different domains dominate based on their density, noise, and size. Diffusion models make these effects more explicit through score decomposition, while in AR models they are implicit in the likelihood objective. We expect our findings to extend to AR models, although deriving equally explicit formulations remains open.
> - Moreover, many modern robotic policies adopt AR–diffusion architectures, i.e. an autoregressive backbone + a diffusion/flow head for action generation (e.g., VLAs like $\pi_0$, WAMs like DreamZero). So in each forward passing, the denoising prediction can be analyzed within our framework.
>
> >W2: Could you propose automated methods for determining optimal data ratios?
>
> Yes! This is also raised by Reviewer Fwek and VzcB. Rather than a single optimum, we observe a **robust effective range** Please check our response to W5 of Reviewer VzcB for the proposed guideline to narrow down the search space.  We hope readers can rely on this to find this range in actual large-scale industrial applications to save cost.
>
> >W3: While ADDA aims to erase domain differences, CFG relies on domain labels. Do these two objectives conflict during joint training, and how do you ensure they don't cancel each other out?
>
> Thanks for the careful observation. Actually what we found is that these two objectives won't conflict, being really stable in joint training, which we think is because the conflicting supervision signal (align/discern) is not from the same pass but being disentangled. The critical metric is the accuracy of the discriminator, which should be converged to ~0.5. This is to ensure that the adversarial training doesn't collapse. We don't start to train the discriminator until after the first 5k steps, as the representation optimization is unstable in the initial stage.
>
>
> >W4: Does the theory still hold in scenarios of extreme data imbalance, such as when only 5-10 real-world trajectories are available?
>
> Yes we believe the theory would still hold in such case. We conduct another series of experiments with the dataset size of 10:3000, and the best performing ratios still locate in the range identified by our framework as shown [here](https://dynamics-humanoid.github.io/cotraining-anonymous/). But it's worthnoting that our analysis doesn't consider the learning dynamics in mini-batch settings, which we acknowledge as a limitation.
>
>
> >W5: How do you quantitatively define the optimal "structured alignment"?
>
> We think that there is not an optimal "structured alignment" but an effective range. In the controlled toy models, we can quantify the results to determine the optimal structured alignment; but generally we argue that such effective alignment is mainly characterized by two properties: alignment and discernibility. Please refer to our response to W1 of reviewer AFgA for a mathematical definition.
>
>
> >W6: To better define the scholarly contribution of this work, it is suggested that the authors include a discussion of some related works.
>
> Thank you for the suggestion! We also notice these good works which is based on the intuition of representation/data alignment. Our work mainly focuses on the mechanisms understanding and also reaches similar conclusions to these works. Firstly, we don't think these works have "addressed" the issues, neither does our work. For example, we compared with the optimal transport technique used in [1], which turns out to be unstable without the offline data pairing. Secondly, our work provide new insight as "structured alignment", which also emphasizes the discernibility of learned representation to adapt the knowledge. It is also a unified view to understand the previous techniques. Due to page limit, we didn't include detailed discussion of related work in the paper. We promise we will include the related work part to dicuss with these works in the next version of the paper.
>
>
> [1] Cheng et al., *Generalizable Domain Adaptation for Sim-and-Real Policy Co-Training*, NeurIPS 2025.

---

> > ### Author Rebuttal · Reviewer_c6Lf · 2026-04-03
> >
> > I still have some concerns that haven’t been addressed, and I hope the author will further elaborate on the similarities and differences between this work and the related work.

---

> > > ### Author Response · Authors · 2026-04-03
> > >
> > > Thank you for your acknowledgement. Below, we provide a more detailed comparison with related approaches to better clarify our contributions and positioning.
> > >
> > > ### **Alignment and discernibility: motivation, mechanism, and space**
> > >
> > > Works such as [1], [3], and [4] share a common motivation that alignment should facilitate cross-domain transfer.
> > > - [1] (OT-based alignment) introduces optimal transport regularization to align latent features in human–robot co-training. Similar to our work, alignment is enforced in the representation space, and empirical gains are observed. **Our contribution complements this line of work by providing a more principled explanation: we show that representation alignment naturally emerges from joint training and is necessary for effective transfer.** Importantly, we further demonstrate that alignment alone is insufficient—maintaining domain discernibility is equally critical due to inherent domain gaps.
> > > - [3] (data filtering via classifier) advances this idea by identifying that certain source-domain samples can be detrimental and should be discarded. Their use of a classifier to separate useful vs. harmful data is conceptually aligned with our notion of domain discernibility. **A key difference is that we jointly learn the classifier and policy, whereas [3] relies on a pre-trained classifier and removes data.** In contrast, we retain all data and instead rely on the model to adaptively reweight their contributions. **These approaches are complementary, and combining them is an interesting direction for future work.** Additionally, [3] performs alignment at the input level (e.g., via keypoint extraction), whereas our focus is on latent representation alignment.
> > > - [4] (data editing via diffusion) performs alignment in data space by transforming source-domain data toward the target domain, akin to approaches such as CycleGAN. From our perspective, such edited data can be interpreted as a form of synthetic domain bridging, similar in spirit to simulation. However, a residual domain gap may still remain. **Our framework is orthogonal to this approach and could be naturally combined, where edited data serves as an additional source domain within co-training.**
> > >
> > > ### **Compositional policy learning perspective**
> > > - [2] (continual/compositional learning) also focuses on the problem of learning from heterogeneous domains, framing the problem through continual adaptation and skill composition.
> > > [2] assumes access to a pre-trained multi-skill policy and performs parameter-space composition via LoRA-based finetuning. **In contrast, our work studies co-training from scratch, where composition occurs implicitly through shared representations, which can be viewed as probabilistic composition in feature space during training.**
> > > - An insightful perspective from [2] is the distinction across parameter, noise, and action spaces, where different levels of alignment and specialization emerge. **This closely relates to our findings: we identify three regimes—over-alignment, disjoint representations, and structured alignment.** Notably, our analysis reveals that co-training in the noise space can induce structured alignment in representation (or parameter) space, offering a complementary and mechanistic explanation of how such compositionality arises.
> > >
> > > ### **Summary**
> > >
> > > Overall, our work differs from prior approaches in that it does not prescribe a specific alignment mechanism, but instead provides a mechanistic understanding of co-training. We identify structured representation alignment—balancing alignment and discernibility—as a key principle, and show how it emerges from training dynamics and governs transfer performance.
> > >
> > > ---
> > > We will summarize these discussion and include it in the related paper of our paper. We hope these clarifications better highlight the novelty and relevance of our contributions, and we respectfully hope this strengthens your assessment of our work.
> > >
> > >
> > > [1] Cheng et al., *Generalizable Domain Adaptation for Sim-and-Real Policy Co-Training*, NeurIPS 2025.
> > >
> > > [2] Liu, T., Li, J., Zheng, Y., Niu, H., Lan, Y., Xu, X. and Zhan, X., Skill Expansion and Composition in Parameter Space. In The Thirteenth International Conference on Learning Representations, 2025
> > >
> > > [3] Pace, M.A., Dan, P., Ning, C., Bhardwaj, A., Du, A., Duan, E.W., Ma, W.C. and Kedia, K., 2025. X-Diffusion: Training Diffusion Policies on Cross-Embodiment Human Demonstrations. arXiv preprint arXiv:2511.04671.
> > >
> > > [4] Niu, H., Chen, Q., Liu, T., Li, J., Zhou, G., Zhang, Y., Hu, J. and Zhan, X., xted: Cross-domain adaptation via diffusion-based trajectory editing. arXiv preprint arXiv:2409.08687.

---

### Official Review · Reviewer_VzcB · 2026-03-12

**Soundness:** 2
**Presentation:** 2
**Significance:** 3
**Originality:** 2
**Overall Recommendation:** 3
**Confidence:** 2

**Summary:**

This paper aims to explore the underlying mechanisms of sim-and-real co-training in generative policies. Using diffusion models as the primary subject of study, and through theoretical derivation of the optimal score function alongside controlled toy experiments, the authors identify two key intrinsic factors that determine the effectiveness of cross-domain knowledge transfer: "structured representation alignment" and "balanced data mixing ratio". The authors further suggest that, when the mixing ratio is appropriately balanced, the learned representations can achieve cross-domain alignment while still preserving domain-specific distinctions. Motivated by this perspective, the paper revisits existing co-training approaches and proposes a combined strategy, CFG-ADDA, which integrates CFG with ADDA. The method is designed to explicitly encourage representation alignment while maintaining domain awareness. Experimental results on three robotic manipulation tasks suggest that the proposed combination can lead to improved performance compared to several baseline variants.

**Compliance With Llm Reviewing Policy:**

Affirmed.

**Final Justification:**

The work lacks sufficient mathematical proof or quantitative verification. Thus, I kept the score unchanged.

**Key Questions For Authors:**

Although the paper explores the role of the balanced mixing ratio $w$ through heuristic analysis, must one still rely on computationally expensive grid searches to find this ratio in actual large-scale industrial applications? Can the authors provide an empirical range based on the data scale ratio between the source and target domains, or a prior estimation of domain gaps, to directly guide practice and narrow the search space for $w$?

**Limitations:**

yes

**Strengths And Weaknesses:**

Strengths:
- This work provides a mechanistic explanation for the underlying mechanisms of sim-and-real co-training to a certain extent, pointing out two key intrinsic factors: "structured representation alignment" and "balanced data mixing ratio".

- It adopts a progressive experimental path, from a highly controlled toy example to sim-and-sim experiments with strictly controlled domain gaps , and finally to sim-and-real real-world validation.

- The paper highlights a core insight: maintaining domain discernibility during cross-domain co-training and transfer learning can improve generalization in the target domain.

Weaknesses:
- The experiments do not include a comparison with the commonly used simulation pre-training + real-world fine-tuning paradigm, which would help clarify the relative advantages of the proposed co-training strategy. In addition, the real-world evaluation is conducted over only 15 episodes. Given the stochasticity of diffusion policies and variability in physical environments, this sample size may lead to high variance in the estimated success rates. Increasing the number of trials (e.g., to around 30 or more) or reporting confidence intervals would help improve the robustness of the results.

- When deriving the optimal score function (Appendices A.2 and A.3), the paper simplifies it into several Gaussian and Softmax terms and relies on assumptions such as "Gaussian concentration in high-dimensional space". However, when dealing with highly non-linear real-world visuomotor streams, whether these assumptions hold true lacks sufficient mathematical proof or quantitative verification.

- The main text provides limited descriptions of the model architecture and training process. To ensure the reproducibility of this work, it is necessary to provide a complete table of hyperparameters, random seed settings, total training steps, data augmentation strategies, and specific network architecture details for the discriminator and encoder.

---

> ### Author Rebuttal · Authors · 2026-03-31
>
> We thank the reviewer for the careful reviewing and valuable feedback! Please see our responses addressing the specific concerns below:
>
> >W1: The experiments do not include a comparison with the commonly used simulation pre-training + real-world fine-tuning paradigm, which would help clarify the relative advantages of the proposed co-training strategy.
>
> We agree this is an important baseline. While prior work shows it is generally less effective than co-training, we include a direct comparison below:
>
> | Task | Pretrain+FT | Co-train |
> |------|-------------|----------|
> | NA   | 0.77        | **0.925** |
> | MC   | 0.215       | **0.495** |
> | MH   | 0.225       | **0.26**  |
>
> Our results are consistent with prior findings [1] (Fig. 4). We will include this comparison in the paper.
>
> >W2: Increasing the number of trials (e.g., to around 30 or more) or reporting confidence intervals would help improve the robustness of the results.
>
> - Thanks for this valuable suggestion, we only ran 15 as we have too many policy variants. Now we increased to 30 episodes per policy checkpoint. The results further strengthen the advantage of our method. Please refer to our response to W1 of Reviewer Fwek.
>
> >W3: When deriving the optimal score function (Appendices A.2 and A.3), the paper simplifies it into several Gaussian and Softmax terms and relies on assumptions such as "Gaussian concentration in high-dimensional space".
>
> Thanks for your careful reminder! But we argue that:
> - **Gaussian distribution assumption**:
>     - Actually the general formulation (Eq.12) of the empirical optimal solution **does not depend on Gaussianity** in Appendix A.1, so the existence of the two intrinsic factors always hold true regardless of the actual mapping.
>     - The softmax term also doesn't rely on the Gaussian form but the exact form of $r_k$, which is $r_k(a^t,t) = \frac{||a^t - \alpha_t a_k||}{\sigma_t \sqrt{d}}$. This is derived for us to get a interpretation of the balancing effect (logit modulation), rather than a constraint.
>     - While Gaussian cannot cover all the cases, it's the default basis for generative model analysis. We believe a more complete proof can be an exciting future work.
>
> - **Gaussian concentration property**:
> Yes we rely on it to support the empirical estimation for the balanced mixing ratio in Appendix A.3.
>     - Firstly, we argue that this property is a well-established **provable** property for Gaussians.
>     - Secondly, this fact actually only influences the curves in Figure.7 of appendix, and the exact form of Eq.17. But the general trend will still hold.
>
> >W4: The main text provides limited descriptions of the model architecture and training process.
>
> Thanks again for your careful reminder! We promise we will include the details in the next version of our paper, and also open-source the complete code for reproducibility. Specifically, our setting details are provided [here](https://dynamics-humanoid.github.io/cotraining-anonymous/) for your interest.
>
>
> >W5: A principled guideline for mixing ratio selection.
>
> This is also raised by Reviewer Fwek and c6Lf. We formally state the guideline here:
> - Given source domain dataset with size of N and target domain dataset with size of M ($M>N$), compute the natural mixing ratio $w_n=\frac{N}{N+M}$, which would be **lower bound** of the range;
> - If $M\gg N$, say $>5$, the **upper bound** could simply be around $w_q=\sqrt{N/M}$; else say $q=0.8$ which is relative portion of real data we want for denoising, compute the upper bound as $w\_q=\frac{N\*q}{\(1-q\)\*M+N\*q}$ (or empirically set the upper bound to $0.5$);
> - Appropiately increase the ratio according to the domain gaps including visual, physics (mainly), embodiment (We don't have a formal way for estimation). Finally do a simple search in the narrowed space $(w_n, w_q)$.
>
> To support this guideline, we additionally conduct another series of experiment with 3 different dataset ratios: 10:3000, 50:500, 50:100. The results [here](https://dynamics-humanoid.github.io/cotraining-anonymous/) indicate that the best performing ratio locates in the range of our guideline.
>
> [1] Wei et al., *Empirical Analysis of Sim-and-Real Co-training of Diffusion Policies*, IROS 2025.

---

> > ### Author Rebuttal · Reviewer_VzcB · 2026-04-03
> >
> > The rebuttal helped, but the core issues remain. Since addressing them would require a substantial update to the paper, I’m leaving my score unchanged.

---

> > > ### Author Response · Authors · 2026-04-03
> > >
> > > We thank the reviewer for the follow-up and are glad that the rebuttal was helpful. However, we are unclear about what specific “core issues” remain unresolved, especially given that the concerns raised in the initial review have been directly addressed.
> > >
> > > To ensure we have not misunderstood, we summarize the original concerns and our corresponding updates:
> > > - **Comparison with simulation pre-training + real-world fine-tuning.** We have added the requested comparisons and updated the results accordingly.
> > > - **Real-world evaluation scale.** We have substantially expanded the evaluation to 30 rollouts per checkpoint, totaling 5(method)x3(task)x3(mixing ratio)x3(checkpoints)x30 rollouts.
> > > - **Assumptions regarding Gaussian distributions.** We clarified that our core derivations do not rely on Gaussian assumptions; and the property of "Gaussian concentration in high-dimensional space" is actually a well-established provable fact.
> > > - **Model architecture and hyperparameters.** We have provided detailed descriptions and included a supplementary webpage for full reproducibility: https://dynamics-humanoid.github.io/cotraining-anonymous/
> > > - **Guidelines for mixing ratio selection.** We have consolidated the guideline and also provided additional experiments for support.
> > >
> > > These updates were made specifically to address the reviewer’s comments. Also these experiments doesn't change the core logic or flow of the paper. Given this, we would greatly appreciate clarification on which aspects are still considered unresolved and why they would require substantial changes to the paper.
> > >
> > > If there are additional expectations beyond the points above, we would be grateful if the reviewer could specify them. Otherwise, we hope the reviewer and AC agree that the main concerns have been substantively addressed in this revision.

---

### Official Review · Reviewer_Fwek · 2026-03-13

**Soundness:** 2
**Presentation:** 3
**Significance:** 2
**Originality:** 2
**Overall Recommendation:** 3
**Confidence:** 4

**Summary:**

The paper investigates the underlying mechanisms of sim-and-real co-training for generative visuomotor policies, specifically focusing on diffusion models. The authors propose that the empirical success of co-training is driven by two intrinsic factors: a "balanced data mixing ratio" and "structured representation alignment". Through theoretical analysis of the diffusion learning objective and a controlled toy MLP experiment , the paper demonstrates that effective knowledge transfer occurs when domain representations are aligned but remain distinguishable. The authors validate this framework through comprehensive sim-and-sim and sim-and-real robotic manipulation experiments using Robosuite , quantifying representation alignment via Wasserstein distance. Finally, the authors analyze existing co-training methods through this lens and propose a combined approach (CFG-ADDA) that explicitly balances representation alignment and domain discernibility, achieving strong empirical results.

**Compliance With Llm Reviewing Policy:**

Affirmed.

**Final Justification:**

The authors' rebuttal has addressed several of my concerns through the expanded evaluation, cross-architecture validation, and clarification on CFG-ADDA's design rationale. However, my concerns about technical novelty remain, as the additional design insights feel more like post-hoc justifications than core methodological contributions, and I therefore maintain my initial score.

**Key Questions For Authors:**

The real-world evaluation utilizes only 15 episodes per task. Given the inherent variance in physical robotic execution, how can you ensure the performance margins between CFG-ADDA and baseline Co-Training are statistically significant?

How robust is the empirical optimal "balanced mixing ratio" across fundamentally different policy architectures, beyond the tested transformer encoder-based diffusion models?

In Figure 5, the correlation between representation alignment and success rate becomes negative under the physics-only domain gap. Could you elaborate on the mechanics of this? Does this imply that enforcing explicit representation alignment is actively detrimental when the underlying physical dynamics are heavily mismatched?

The proposed CFG-ADDA approach is a static combination of two existing techniques. Did you experiment with dynamically adjusting the CFG guidance scale or the adversarial loss weighting during training based on real-time measurements of representation alignment?

**Limitations:**

yes

**Strengths And Weaknesses:**

Strength:

The authors successfully derive an analytical optimal score function for diffusion models under a co-training objective. Furthermore, the empirical setup thoughtfully isolates the effects of domain gaps by explicitly constructing visual-only, physics-only, and visual-physics simulation environments.

Weakness:

- The real-world policy evaluation is conducted on a very small sample size, using only 15 episodes per task. This is insufficient to draw statistically significant conclusions regarding the superiority of the proposed CFG-ADDA method, especially when baseline comparisons have narrow margins. The authors claim to define a range of "balanced" mixing ratios via heuristic analysis , but the main text relies entirely on a manual, empirical grid search to find optimal performance. The lack of a principled, a priori method in the main text to compute this ratio for new datasets severely limits the soundness of the framework.


- The actionable algorithmic contribution (CFG-ADDA)  is a straightforward concatenation of one-hot domain embeddings (CFG) with a standard adversarial discriminator (ADDA). While the theoretical motivation for combining them is sound, the method itself lacks technical novelty and serves merely as an ensemble of two off-the-shelf techniques.

---

> ### Author Rebuttal · Authors · 2026-03-31
>
> We thank the reviewer for the useful feedback, which is aimed to further strengthen the soundness of our contribution. Please see our responses addressing the specific concerns below:
> > W1: The real-world evaluation uses only 15 episodes per task, which is insufficient for statistical significance.
>
> - We agree that larger sample size will be more convincing to support the conclusions, which is also raised by Reviewer VzcB. We increased the evaluation to **30 episodes per policy**. The updated results below consistently support the superiority of **CFG-ADDA**, with improved margins across tasks:
>
> | Method | NA(.016) | NA(.1) | NA(.3) | MC(.016) | MC(.1) | MC(.3) | MH(.016) | MH(.1) | MH(.3) | Avg |
> |--------|----------|--------|--------|----------|--------|--------|----------|--------|--------|-----|
> | Real   | 11 | 11 | 11 | 8 | 8 | 8 | 7 | 7 | 7 | 8.6 |
> | CoT    | **17** | 11 | 16 | **16** | 9 | 7 | 8 | **13** | 7 | 15.3 |
> | OT     | 15 | **17** | 11 | 8 | 15 | **15** | **11** | 9 | 4 | 14.3 |
> | ADDA   | 13 | 13 | **15** | 6 | **14** | 11 | 10 | **14** | 7 | 14.3 |
> | CFG    | **15** | 14 | 11 | 6 | **17** | 14 | 8 | **14** | 10 | 15.3 |
> | C+A    | **23** | 15 | 18 | 11 | **22** | 17 | **18** | 15 | 8 | **21** |
>
> - Additionally, we report **sim-and-sim results with 200 rollouts per policy**, ensuring statistical reliability. the full results are provided [here](https://dynamics-humanoid.github.io/cotraining-anonymous/), Figure.6 in the paper is distilled down from this.
> - We appreciate that the reviewer also notice that the baselines have narrow gains. Actually these techniques haven't been broadly accepted in robotics co-training. Prior work reports mixed stability and gains. For example the "ADDA" baseline, in [1] the authors find certain improvement while in [2] the authors find it not stable, suggesting this direction remains underexplored.
>
> > W2: Lack of a principled automated method to compute the mixing ratios.
>
> This is a valuable advice! A coarse guideline is actually in the Appendix A.3 of the main paper. Please refer to our response to W5 of Reviewer VzcB.
>
> > W3: How robust is the optimal “balanced mixing ratio” across architectures, beyond the tested transformer encoder-based diffusion models?
>
> We find that this range is robust across architectures. We validate this using **ResNet+U-Net and encoder-free transformer** backbones under identical settings on the same task suites:
>
> || | w=0 | w=0.005 | w=0.016 | w=0.1 | w=0.3 | w=0.5 | w=0.8 | w=1.0 |
> |--------------|--------------|-----|---------|---------|-------|-------|-------|-------|-------|
> | NutAssembly  | Encoder-free | 0   | 0.79    | 0.86    | **0.88**  | 0.865 | 0.86  | 0.82  | 0.765 |
> |              | U-net        | 0   | 0.86    | **0.95**    | 0.905 | 0.875 | 0.885 | 0.85  | 0.82  |
> | MugCleanup   | Encoder-free | 0   | 0.255   | 0.375   | 0.4   | **0.415** | 0.3   | 0.255 | 0.23  |
> |              | U-net        | 0   | 0.38    | 0.395   | **0.4**   | 0.355 | 0.325 | 0.295 | 0.295 |
> | MugHang      | Encoder-free | 0   | 0.14    | 0.25    | 0.255 | **0.275** | 0.25  | 0.26  | 0.245 |
> |              | U-net        | 0   | 0.25    | **0.32**    | 0.315 | 0.295 | 0.28  | 0.265 | 0.225 |
>
> Across all tasks, the best performance consistently lies within a **narrow range (in [0.016, 0.3])**, indicating robustness beyond the architecture used in the main paper.
>
> >W4: In Figure 5, the correlation between representation alignment and success rate becomes negative under the physics-only domain gap. Could you elaborate on the mechanics of this?
>
> In the physics-only gap, input distribution gaps are much smaller, leading to low representation distance and **reduced domain discriminability**, which drops to around ~90% in our setting. This places representations in a regime between overlap and structured alignment, where over-alignment harms performance. The negative correlation suggests that **alignment without sufficient discernibility is detrimental when underlying physics differ**.
>
> > W5: Did you experiment with dynamically adjusting the CFG guidance scale or the adversarial loss weighting during training based on real-time measurements of representation alignment?
>
> Yes! We experiment with different CFG scale during inference. The results can be found [here](https://dynamics-humanoid.github.io/cotraining-anonymous/). We haven't tried other dynamic training strategy, which could have a large design space for future work, but our training has dynamic schedule designed with the same spirit: we only start to train our discriminator after the starting 5k steps, as the representation is pretty unstable at the start of training. We mainly aim at understanding co-training in this project.
>
> [1]Cai, Xiongyi, et al. "In-N-On: Scaling Egocentric Manipulation with in-the-wild and on-task Data." arXiv:2511.15704 (2025).
> [2]Yuan, Chengbo, et al. "Motiontrans: Human vr data enable motion-level learning for robotic manipulation policies." arXiv:2509.17759 (2025).

---

> > ### Author Rebuttal · Reviewer_Fwek · 2026-04-04
> >
> > Thank you for the detailed rebuttal. The expanded evaluation (30 episodes + 200 sim-and-sim rollouts), the cross-architecture validation, and the mixing ratio guideline have substantially strengthened the empirical foundation of the work. I consider W1, Q2, Q3, and Q4 from my original review to be adequately addressed.
> >
> > My remaining consideration is W2 regarding the technical novelty of CFG-ADDA. I understand that the paper positions itself primarily as a mechanistic study rather than a method contribution, and I appreciate this framing. However, given that CFG-ADDA is presented as the actionable outcome of the theoretical insights, I would like the authors to clarify: what prevents a practitioner from arriving at this exact combination without the theoretical analysis? In other words, what specific design decision in CFG-ADDA is informed by the structured alignment insight that would not be obvious from simply trying existing techniques?
> >
> > If the authors can articulate this connection more concretely, I am willing to raise my score.

---

> > > ### Author Response · Authors · 2026-04-05
> > >
> > > Thanks for your thoughtful follow-up and for recognizing our framing. We try to articulate clearly about the motivation and technical contribution shaped from the theoretical analysis:
> > >
> > > ---
> > >
> > > ### **Review: How does our understanding motivate the method design?**
> > >
> > > **Restatement**: Our analysis leads to the concept of **structured representation alignment**, which highlights a key principle: effective co-training requires *both* cross-domain alignment *and* domain discernibility. These two factors are often treated as competing objectives, and prior methods typically emphasize only one side (e.g., alignment via OT/ADDA, or discernibility via CFG). This perspective reframes existing techniques under a unified lens and, more importantly, suggests that **balancing these two forces is necessary rather than optional**. This directly motivates combining CFG with ADDA.
> > >
> > > ---
> > > ### **Q: What prevents a practitioner from arriving at this combination without our analysis?**
> > >
> > > We agree that, in principle, such a combination could be discovered empirically. However, our contribution lies in **making this combination *motivated, targeted, and non-trivial*** rather than incidental:
> > >
> > > **Perspective 1: It's possible but might be hard for a practioner to possess the motivation to try out this combination, and while our undertanding narrows down the search space.** We totally agree that it is possible that people might try this combination in somedays. But there are some obstacles that hinder this process while we accelerate this process.
> > > - **Lack of motivation to combine these two "seemingly contraditive" objectives.** People usually consider the two methods as two contradictive supervision signals and don't have the strong motivation to combine them It might hinder this discovery process if people just randomly explore, though still being possible.
> > > For example, in [1] the authors explicitly tried out CFG and also representation alignment methods like MMD and ADDA. But they treat them as two completely different methods, and didn't try to combine them. Our analysis shows that this “conflict” is only superficial, and that **the two mechanisms operate at different levels and can be complementary when properly balanced**, thereby substantially narrowing the search space. (A analogy is like model-free RL and model-based RL, our understanding acts as a model and improves the sample-efficiency on the method discovery.)
> > >
> > >
> > > **Perspective 2: It's more than a naive static combination, while it brings new discovery.**
> > > - **The design of "where to apply alignment" in CFG-ADDA is only possible with our understanding.** We apply alignment regularization only on the observation representation $z$, while **explicitly preserving domain information in the conditioning variables**. A naive approach, e.g., aligning the full condition space $(z, c)$ with learnable $c$ would collapse domain distinctions and lead to conflicting training signals—this concern is also reflected in reviewer feedback (e.g., c6Lf).
> > > - **This combination brings unexpected new discovery and understanding of old technique.** In our experiments of dynamic adjusting the guidance scale of CFG, our results show that setting appropriate negative guidance value can improve the performance. However, previous studies [1][2][3] using CFG in robotic policies only consider non-negative guidance scale, which brings new understanding of the coefficient $\lambda$ in CFG -- as only the environment labels are dropped in $s_\theta(a,o,\varnothing,t)$, it actually represents the average log-probability gradient direction in all domains.
> > > So $\lambda$ can be viewed as a more flexible control variable to transfer the "averaged knowledge" during inference.
> > >
> > >
> > > These design choices are **direct consequences of the structured alignment perspective**, and are unlikely to emerge from unguided trial-and-error. We hope this clarifies that CFG-ADDA is not merely an ad hoc combination, but a concrete instantiation of the underlying mechanism we uncover. Finally, we mainly hope this could be a starting point to motivate new paradigm design in the future, so we also plainly name the proposed technique as "CFG-ADDA".
> > >
> > > ---
> > > Overall we hope this can articulate the connection and scholarly contribution more clearly. We will incorporate this discussion into the revised version for clarity.
> > >
> > > [1] Wei, Adam, et al. "Empirical analysis of sim-and-real cotraining of diffusion policies for planar pushing from pixels." 2025 IEEE/RSJ International Conference on Intelligent Robots and Systems (IROS). IEEE, 2025.
> > >
> > > [2] Chi, Cheng, et al. "Diffusion policy: Visuomotor policy learning via action diffusion." The International Journal of Robotics Research 44.10-11 (2025): 1684-1704.
> > >
> > > [3] Reuss, Moritz, et al. "Goal-conditioned imitation learning using score-based diffusion policies." arXiv preprint arXiv:2304.02532 (2023).

---

### Decision · Program_Chairs · 2026-04-30

**Decision:**

Accept (regular)

**Comment:**

The paper receives mixed (but improved) reviews after the rebuttal, with multiple reviewers acknowledging that key requested additions were provided. In particular, additional experiments on the theoretical regimes and clarifications on related work are viewed as meaningful improvements.

Two reviewers still maintain weak reject positions.
- Reviewer VzcB has three broad concerns about empirical significance, theoretical assumptions, and reproducibility. This is in contrast to Reviewer c6Lf who mentions in the final justification that the "rebuttal has improved my confidence in the paper’s technical contribution and positioning" and Reviewer AFgA who mentions that the contribution can lead to "better formalization of structured representation alignment" and Reviewer AFgA also praises the additional experiments promised by the authors. Concerning "reproducibility", the authors provided a list of improvements that address the concerns and reproducibility is not listed as a major concern by the other reviewers. The final justification of that reviewer mentions briefly "the work lacks sufficient mathematical proof or quantitative verification" but the concerns do not seem clearly substantiated.
- Reviewer Fwek keeps a concern about technical novelty of CFG-ADDA in their final justification: "my concerns about technical novelty remain, as the additional design insights feel more like post-hoc justifications than core methodological contributions". The main reason for the lack of novelty from Reviewer Fwek was related to the fact that the approach could be seen as an "ensemble of two off-the-shelf techniques". Reviewer c6Lf views the originality as a strength in the review and Reviewer AFgA mentions "Structured representation alignment is an interesting concept, and the theoretical argument for its importance is convincing".

Overall, the rebuttal appears to have addressed specific actionable requests that lead to two reviewers having "weak accept" scores, while two have "weak reject" scores. The reviews are mixed with differing opinions to judge significance or originality. While the paper could benefit from a more thorough investigation on some aspects and/or additional theoretical results, the contribution is overall sound and can likely be useful to the research community. The paper is at a weak accept level.